# Simple and Scalable Strategies to Continually Pre-train Large Language Models

**Adam Ibrahim**[*†⊙]                                           *research@adamibrahim.fr*
**Benjamin Thérien**[*†⊙]                                *benjamin.therien@mila.quebec*
**Kshitij Gupta**[*†⊙]                                       *kshitij.gupta@mila.quebec*
**Mats L. Richter** [†⊙]                                    *mats.richter@mila.quebec*
**Quentin Anthony** [◇†⊙]                                    *qubitquentin@gmail.com*
**Timothée Lesort** [†⊙]                                          *t.lesort@gmail.com*
**Eugene Belilovsky** [‡⊙]                          *eugene.belilovsky@concordia.ca*
**Irina Rish** [†⊙]                                          *irina.rish@mila.quebec*

*Department of Computer Science and Operation Research,*
*Université de Montréal, Montréal, Canada †*

*Department of Computer Science and Software Engineering,*
*Concordia University, Montréal, Canada ‡*

*Mila, Montréal, Canada ⊙*

*EleutherAI ◇*

**Reviewed on OpenReview:** *https://openreview.net/forum?id=DimPeeCxKO*

## Abstract

Large language models (LLMs) are routinely pre-trained on billions of tokens, only to start the process over again once new data becomes available. A much more efficient solution is to continually pre-train these models—saving significant compute compared to re-training. However, the distribution shift induced by new data typically results in degraded performance on previous data or poor adaptation to the new data. In this work, we show that a simple and scalable combination of learning rate (LR) re-warming, LR re-decaying, and replay of previous data is sufficient to match the performance of fully re-training from scratch on all available data, as measured by the final loss and the average score on several language model (LM) evaluation benchmarks. Specifically, we show this for a weak but realistic distribution shift between two commonly used LLM pre-training datasets (English→English) and a stronger distribution shift (English→German) at the 405M parameter model scale with large dataset sizes (hundreds of billions of tokens). Selecting the weak but realistic shift for larger-scale experiments, we also find that our continual learning strategies match the re-training baseline for a 10B parameter LLM. Our results demonstrate that autoregressive transformer-based LLMs can be successfully updated via simple and scalable continual learning strategies, matching the re-training baseline using only a fraction of the compute. Finally, inspired by previous work, we propose alternatives to the cosine learning rate schedule that help circumvent forgetting induced by LR re-warming and that are not bound to a fixed token budget.

## 1 Introduction

Over the past few years, large pre-trained models have enabled massive performance improvements in language modeling (Brown et al., 2020; Zhao et al., 2023), visual understanding (Radford et al., 2021; Alayrac et al., 2022; Kirillov et al., 2023), text-to-image generation (Rombach et al., 2022; Pernias et al., 2024), and text-to-video generation (Brooks et al., 2024)—to name a few. Large language models (LLMs) are at the

center of all these improvements, providing an intuitive means for humans to interface with machine learning algorithms through language.

While LLMs are the cornerstone of current generative AI technology, they are expensive to train and keep up to date. However, as new and higher-quality datasets continue to become available (Gao et al., 2020; Soboleva et al., 2023; Computer, 2023; Soldaini et al., 2024), organizations will need to update their models to stay abreast of the competition. Currently, LLMs are re-trained on a combination of old and newly collected data. Existing works aim to reduce these training costs by enabling low-cost hyperparameter optimization (Yang et al., 2022) or providing guidelines for maximizing performance under a given compute budget (Hoffmann et al., 2022). However, these works assume that models will be *trained from random initialization*, raising the following question: Should practitioners always combine existing datasets and *train from random initialization* to obtain the best performance? Doing so for every update of the models quickly becomes expensive.

To avoid complete re-training, we explore simple and scalable continual learning strategies for continuing to pre-train LLMs (up to 10B parameters) on large amounts of new data (200B+ tokens). We refer to our setting as "continual pre-training" and highlight that it is *distinct* from existing settings in the literature (Gururangan et al., 2020; Ke et al., 2022; Scialom et al., 2022; Xie et al., 2023) due to the large amount of incoming data we consider. In this work, we do not intend to improve on the performance of models trained from a random initialization on all of the available data. Instead, we consider models trained on the union of existing datasets as baselines whose performance we seek to match using a combination of continual learning strategies at scale.

Naively continuing to train the model on new data, however, tends to lead to performance far below re-training on all available data, often due to 1) poor adaptation (failure to optimize the new dataset) or 2) catastrophic forgetting (significant capability loss on the previous dataset). Firstly, the question of adaptation is central to our setting as training on large datasets is costly. One would presumably not choose to spend considerable computational resources training on a new dataset only to minimally adapt to it. However, most performant open-source LLMs (Touvron et al., 2023a;b; Jiang et al., 2023; Gemma Team et al., 2024) decay their learning rate to a small value by the end of training. We hypothesize, therefore, that the learning rate must be re-increased and re-decayed to improve adaptation per compute spent when training on a new dataset. We note that this has not been thoroughly studied in the continual learning literature. Secondly, catastrophic forgetting is a key difficulty to overcome if one is to realize the full potential of continual pre-training. Adapting to hundreds of billions of new tokens is important, but it must not come at the cost of erasing most existing knowledge in the LLM. Recent work (Scialom et al., 2022) shows, in an LLM fine-tuning setting, that replaying previous data (as little as 1%) is sufficient to mitigate forgetting to a large extent. While continually pre-training on large amounts of new data will almost surely lead to more forgetting than fine-tuning, we hypothesize that an appropriate amount of replay could mitigate forgetting—even in our setting. Moreover, recent works show that pre-training (Cossu et al., 2022; Ramasesh et al., 2022; Mehta et al., 2023) and increasing model size (Mirzadeh et al., 2022) both help to reduce the effects of forgetting. We, therefore, expect the trend of increasing language model capacity and pre-training dataset size in tandem (Kaplan et al., 2020; Hoffmann et al., 2022; Touvron et al., 2023b) will yield models increasingly capable of continual learning (Scialom et al., 2022), suggesting that our experimental results should only improve with models scale.

Given the great potential for continual learning to considerably reduce costs associated with re-training models and the potential for LLMs to be strong continual learners, we ask ourselves the following question: *when simple and scalable continual learning techniques are applied, what is the performance difference between continually pre-trained LLMs relative to LLMs pre-trained from random initialization on the union of all data?* To answer this question, we conduct a large-scale empirical study of continual learning techniques for LLM pre-training. Our empirical evaluation spans large (10B parameters) and small (405M parameters) decoder-only transformer models as well as weak (English → English) and stronger (English → German) distribution shifts. Our main contributions can be summarized as follows:

1. We establish the effect of learning rate re-warming and re-decaying for decoder-only transformer-based LLMs pre-trained using a cosine schedule, showing that re-warming and re-decaying is necessary for adaptation during continual pre-training.

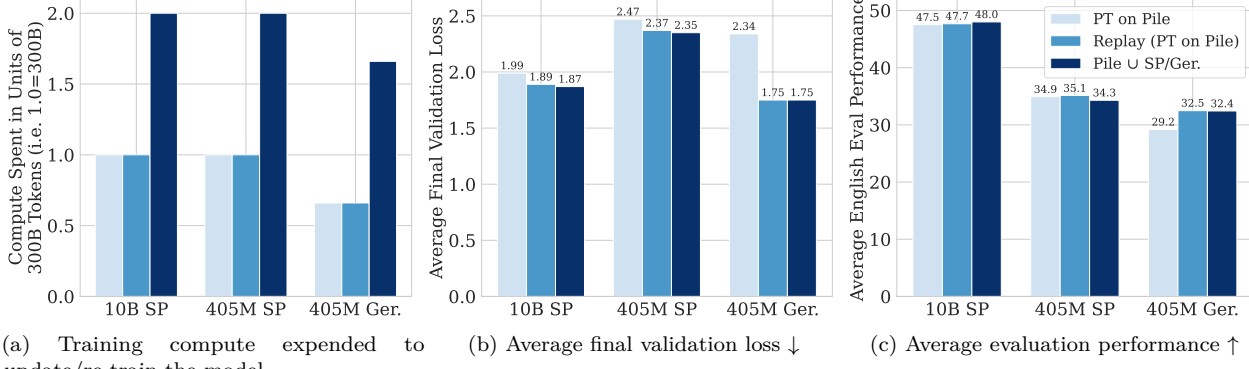

(a) Training compute expended to update/re-train the model

(b) Average final validation loss ↓

(c) Average evaluation performance ↑

Figure 1: **Continual pre-training decreases computational costs of updating the model while maintaining similar final validation and average evaluation performance.** We report results for the Pile ∪ SlimPajama(SP)/German(Ger.) baseline model trained on the union of both datasets which we consider to be an upper bound on performance. We also report performance for two continually pre-trained models. "PT on Pile" starts from a pre-trained Pile checkpoint and only uses learning rate re-warming and re-decaying, while "Replay (PT on Pile)" re-warms the learning rate, re-decays it, and uses 5% replay for SlimPajama and 25% replay for German. We observe that the combination of LR re-warming, re-decaying, and replay allows our continually pre-trained model to attain similar average performance to the baseline model while requiring substantially less compute. We note that this setting assumes that a pre-trained model is available (e.g., via HuggingFace hub or an in-house model designed to be continually pre-trained).

2. We establish the effect of replaying previous data while keeping compute constant across two distribution shifts and many replay percentages. We find that, even when updating decoder-only transformer-based LLMs on hundreds of billions of new tokens, it is possible to significantly mitigate forgetting with an appropriate amount of replay.

3. We demonstrate, across two model sizes and distribution shifts, that a simple and scalable combination of LR re-warming, LR re-decaying, and compute-equivalent replay allows continually pre-trained decoder-only transformer-based LLMs to attain similar performance on average to models re-trained on the union of all data while using significantly less compute.

4. We propose infinite learning rate schedules (schedules allowing smooth transition across datasets) for the continual pre-training of LLMs as a promising way to circumvent optimization difficulties associated with learning rate re-warming.

Our code is available at `https://github.com/EleutherAI/gpt-neox` through pull requests 1194 and 1200. Model checkpoints throughout continual pre-training for most of our models are available at `https://huggingface.co/collections/cerc-aai/continual-pre-training-661f4af4379b82d9617a9401`. A preliminary version of this work was made available as an ICML 2023 workshop paper in (Gupta et al., 2023).

## 2 Main Findings and Takeaways and Examples our Method's Practicality

Our experimental results assume that continually pre-trained LLMs undergo two or more pre-training phases sequentially. That is, our results apply to situations where a continually pre-trained LLM is randomly initialized and pre-trained on datasets $\mathcal{D}_0, \mathcal{D}_1, \ldots, \mathcal{D}_{N-1}$ in sequence where $N \geq 2$ and $tokens(\mathcal{D}_i) \geq 100B$. We note that this includes situations where the LLM in question is an open-source model (Touvron et al., 2023a;b; Jiang et al., 2023; Gemma Team et al., 2024) which has already been pre-trained on $\mathcal{D}_0$ and situations where organizations may wish to train an initial LLM with the intention of continually pre-training it on new data. The new data may be similar to the previous data, corresponding to a weak distribution shift (e.g., the latest web-scrape of different domains), or quite different from previous data, corresponding to a strong

distribution shift (e.g., data from a completely new language). Our experimental evaluation accounts for these difficulties, finding that appropriately applying LR re-warming, LR re-decaying, and replay is sufficient to match the performance of re-training across weak and strong distribution shifts and two model sizes (see Fig. 1). To make our findings as accessible to the community as possible, we now provide *Rules of thumb* for applying our findings:

---

**Rules of thumb for continual pre-training**

**Caveat**—The following guidelines are written to the best of our *current knowledge.*

**Learning rate schedule:**

- If the learning rate was cosine-decayed from a large value $\eta_{max}$ to a small value $\eta_{min}$ during pre-training on the initial dataset, the following guidelines can help to continually pre-train your model:

  - Re-warming and re-decaying the learning rate from $\mathcal{O}(\eta_{max})$ to $\mathcal{O}(\eta_{min})$ improves adaptation to a new dataset, e.g. compared to continuing from small learning rates $\mathcal{O}(\eta_{min})$.
  - Decreasing the schedule's maximum learning rate can help reduce forgetting, whereas increasing it can improve adaptation.

- Infinite LR schedules are promising alternatives to cosine decay schedules. They transition into a high constant learning rate across tasks, helping prevent optimization-related forgetting by avoiding re-warming the LR between tasks. They also avoid committing to a specific budget of tokens as a final exponential decay can be used to train the model to convergence at any point during training.

**Replay:**

- We find that even small amounts of replay are good at mitigating forgetting. We recommend experimenting with different replay fractions since relative differences between them appear very early during training. For example, one may experiment with different replay fractions for a limited token budget, using evaluations relevant to their use case, to find a sweet spot between adapting to the new data and mitigating performance loss due to the distribution shift.

---

**Recent works employing our techniques**   Two notable recent works (Glorioso et al., 2024; DeepSeek-AI et al., 2024) have successfully applied combinations of the techniques proposed herein to continually pre-train LLMs at scale, providing further evidence of their efficacy. Glorioso et al. (2024) apply LR re-warming, LR re-decaying, and 60% replay in the context of a decay phase over 50B tokens of high-quality data, applied after their initial pre-training phase. The authors observe improvements in their model's performance without suffering from catastrophic forgetting. DeepSeek-AI et al. (2024) select a non-decayed checkpoint from the initial pre-training phase to ensure a smooth LR transition into continual pre-training (e.g., as suggested in Figure 9), use a decay, and use 30% replay of pre-training data, to continually pre-train DeepSeek-V2 (DeepSeek-AI, 2024) on 6T tokens. The resulting model significantly improves its code generation abilities, while retaining most of its natural language generation abilities. Together, these works highlight the generality the techniques we propose herein: applying the appropriate combinations work to continually pre-train LLMs on small and large continual pre-training datasets (e.g., 50B and 6000B, respectively) and for architectures beyond the dense transformer (e.g., hybrid SSM-transformers and sparse Mixture of Experts models, respectively).

# 3 Related Work

## 3.1 Continual learning

Continual learning (CL) approaches aim to learn from an evolving data distribution, adapting to novel data while retaining knowledge gathered through prior training (French, 1999; Rolnick et al., 2019; Caccia et al., 2020; Lesort et al., 2021). The key challenge of continual learning is to avoid forgetting past information, while also adapting to novel information. This trade-off is known as the rigidity-plasticity dilemma (Mermillod et al., 2013; Ostapenko et al., 2019; Riemer et al., 2019).

CL approaches are convenient even in small-scale settings to avoid re-training from scratch or to bridge the data availability issue (Smith et al., 2021). However, at scale, CL is more than a convenience; it may be necessary to process huge amounts of continually gathered data. The recent increase in training scale, most notably for LLMs (Scao et al., 2022; Brown et al., 2020; Zhao et al., 2023), offers new opportunities for CL to reduce the cost of re-training and increase efficiency for memory, computing, and storage (Prabhu et al., 2023; Aljundi et al., 2019; Harun et al., 2023a; Veniat et al., 2021; Harun et al., 2023b). Just as federated learning can enable the sharing of compute and data between different agents co-located in space (McMahan et al., 2017; Reddi et al., 2021; Douillard et al., 2023; Ryabinin et al., 2021), continual learning allows the sharing of compute and data progressively through time and could be a useful tool for large-scale training.

Recent work shows that optimizers such as SGD and Adam have interesting knowledge retention properties in DNNs that could be beneficial at scale for CL (Lesort et al., 2023) and that just a small amount of replay could be sufficient to boost knowledge accumulation (Scialom et al., 2022). In this work, we want to benefit from the efficiency of those approaches in the context of large language models pretraining and boost them with the right learning rate scheduling and replay policy.

## 3.2 Pre-training, Model Scale, and Continual Learning

Several existing works evaluate the impact of pre-training and model scale on continual learning. Cossu et al. (2022) investigate pre-training scenarios for language and vision. They find that unsupervised and self-supervised pre-training plays a fundamental role in mitigating forgetting, while supervision hurts performance. Similarly, Mehta et al. (2023) find that pre-trained models forget less than randomly initialized models, due to their weights lying in flatter regions of the loss landscape. They also find that larger models forget less which is connected to the findings of Ramasesh et al. (2022); Mirzadeh et al. (2022). The former finds that pre-trained models forget less as they are scaled up, suggesting that it may be due to the hidden representations growing more orthogonal with scale. The latter finds that wider neural networks forget less compared to their parameter-equivalent deeper counterparts. Hernandez et al. (2021) establish scaling laws for transfer: equations that can predict the performance of a neural network on a new task as a function of its parameter count and pre-training dataset size. The authors find that this positive transfer consistently improves as the parameter count increases. Finally, Scialom et al. (2022) show that autoregressive LLMs have a strong ability to learn continually which they hypothesize is related to their pre-training objective.

## 3.3 Domain Adaptive Continual Pre-training (DACPT)

Existing work considers Domain Adaptive Continual Pre-training (DACPT), a setting where a series of unlabelled domains become available to the LM sequentially and practitioners wish to train on each domain in a self-supervised fashion while retaining performance across each of them. While the objective is similar to our own, we consider general-purpose pre-training datasets that mix many domains as opposed to domain-specific datasets. Ke et al. (2022) assume data from previous domains is not available when training on new domains and develop a new technique for this setting which involves an importance mask of parameters for all previous tasks to prevent forgetting when pre-training with a masked language modeling (MLM) objective. Gururangan et al. (2020) investigated domain and task adaptive pre-training of RoBERTa (also MLM) and contributed a sample selection strategy for efficient continual pre-training. Similarly, Xie et al. (2023) also propose a data selection strategy that reduces the computational cost of continual pre-training (shown for autoregressive LMs). Qin et al. (2023) investigate re-cycling fine-tuned adapter layers of previous base LMs

as the initialization of new adapters for adapting continually updated versions of the base LM to specific tasks. Recently, Wu et al. (2024) proposed LLaMA Pro, a method for the continual pre-training of LLMs that enables learning new tasks without forgetting previous knowledge. However, unlike our work which considers adapting all existing weights, LLaMA Pro requires growing the size of the model for each new update and only adjusting the new weights.

### 3.4 Continual Learning for LMs Applied to Specific Domains

Several related works apply continual pre-training to specific tasks and domains (Sun et al., 2020; Jang et al., 2022a;b; Gong et al., 2022; Zan et al., 2022; Yadav et al., 2023a; Ma et al., 2023; Yang et al., 2024). While these works also utilize continual pre-training techniques, they differ from our work by focusing on particular domains instead of general pre-training techniques, on smaller-scale datasets $< 10B$ tokens with smaller models. The only existing work that approaches our dataset scale is (Gogoulou et al., 2023), which explores continual autoregressive language modeling across English, Danish, Icelandic, and Norwegian datasets (73B each). While they do not use replay they do re-warm and re-decay the learning rate. The only existing work that approaches our model scale is (Yang et al., 2024). They continually pre-train and instruction tune LLaMA2 on small-scale academic plant science data. This concurrent work uses a very similar continual learning setup to the one we propose: replay, LR re-warming, and LR re-decaying. While, unlike our work, they *do not* build a controlled experimental framework to systematically evaluate the validity of these approaches for continual pre-training, it is nice to see further experimental evidence validating our approach.

### 3.5 Learning Rate Schedules

Several studies have examined the impact of different learning rate (LR) schedules on the training stability and final performance of neural networks. Goyal et al. (2018) found that a gradual warm-up of LR early on in training can help overcome optimization challenges, particularly with large mini-batch sizes. Additionally, Popel & Bojar (2018) emphasized the importance of a warm-up stage when training Post-LN Transformers. On the other hand, Xiong et al. (2020) discovered that Pre-LN Transformers are more stable and may not require a warm-up stage. You et al. (2019) explored the role of the LR decay and found that a large initial LR prevents the network from memorizing noisy data, whereas a small LR helps learn complex patterns. Kaplan et al. (2020) explored LR schedules for pre-training Large Language Models (LLMs) and found that schedule choice did not significantly impact performance. Correcting this erroneous finding, Hoffmann et al. (2022) found that the LR schedule does play an important role. Hoffmann et al. (2022) and Rae et al. (2021) established best practices for using a cosine schedule when pre-training LLMs, which have become widely adopted. In contrast, Raffel et al. (2023) and Zhai et al. (2022) explore LR schedules that follow the inverse square root decay for large-scale pre-training. Raffel et al. (2023) utilized an inverse square root decay for training LLMs, allowing flexibility in adjusting the number of training steps. In Zhai et al. (2022), authors use these schedules referred to as "infinite learning rate schedules" to train vision transformers. These schedules enable indefinite training and the evaluation of multiple training durations in a single run. We note that our proposed infinite learning rate schedules for LLMs (Sec. 7.4) are inspired by this idea.

## 4 Background & Methodology

In this section, we provide appropriate background and methodology as it relates to continual pre-training in the context of LLMs.

### 4.1 Linear Warmup and Cosine Decay Schedule

Hoffmann et al. (2022) and Rae et al. (2021) established best practices for using a cosine schedule when pre-training LLMs. Specifically, they recommend starting with a linear warmup phase and decaying the learning rate to $10\times$ its maximum value such that the end of the cosine cycle is set to match the number of tokens. While the linear warmup duration differs, most works have a duration between $0.1\%$ and $0.5\%$ of training steps (Zhao et al., 2023). Given that many popular open-source models (Touvron et al., 2023b;a; Almazrouei et al., 2023) follow this learning rate schedule recipe, it is critical to understand its nuances for

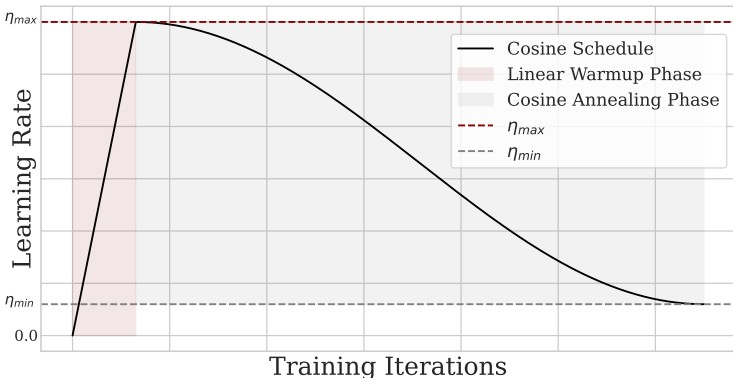

Figure 2: **Linear warmup and cosine annealing schedule.** For illustration purposes, the schedule uses linear warmup for 10% of training iterations. However, most works have a duration between 0.1% and 0.5% of training steps (Zhao et al., 2023).

continually pre-training such models. The schedule first linearly increases the learning rate over $T_{warmup}$ timesteps, or equivalently until some timestep $t_{ann} = T_{warmup}$:

$$\eta_t = \eta_{max} \cdot \frac{t}{T_{warmup}} \tag{1}$$

where $\eta_t$ is the value of the learning rate at iteration $t$, and $\eta_{max}$ is the maximum learning rate. The schedule then transitions into a cosine annealing phase over $T_{ann}$ timesteps, equivalently until some timestep $t_{end} = T_{ann} + t_{ann}$:

$$\eta_t = \eta_{min} + \frac{(\eta_{max} - \eta_{min})}{2} \cdot \left( \cos \left( \pi \cdot \frac{t - t_{ann}}{t_{end} - t_{ann}} \right) + 1 \right) \tag{2}$$

where $\eta_{max}$ is the maximum learning rate and $\eta_{min}$ is the minimum learning rate. Fig. 2 illustrates these two phases.

## 4.2 Compute-equivalent Replay

In many of our experiments, we compare models trained with replay to models trained without it. When making such comparisons, we keep the amount of compute constant for training both models. That is, we correspondingly reduce the number of tokens seen from the new dataset to accommodate the additional tokens seen from the replay buffer. We refer to this use of replay as *compute-equivalent replay*. For instance, suppose datasets $\mathcal{D}_0$ and $\mathcal{D}_1$ each contain 100B tokens. We wish to compare model (a) trained sequentially on $\mathcal{D}_0$ and $\mathcal{D}_1$ to model (b) trained sequentially on $\mathcal{D}_0$ and $\mathcal{D}_1$ with 5% compute equivalent replay. Model (a) will see all tokens from both datasets for a total of 200B unique tokens. Model (b) will see 100B unique tokens of $\mathcal{D}_0$ and 95B unique tokens of $\mathcal{D}_1$ plus 5B replayed tokens from $\mathcal{D}_0$ for a total of 200B tokens. In this way, both compared models expend the same amount of compute.

For instance, in our settings that span only two datasets $(\mathcal{D}_0, \mathcal{D}_1)$, we use replay of data from $\mathcal{D}_0$ when training on $\mathcal{D}_1$. We replay the data in the order it was seen when pretraining on $\mathcal{D}_0$, as we did not observe noticeable differences when reshuffling the replay data in preliminary experiments. The use of methods for selecting replay samples is left as future work. We refer to models using replay as "$\mathcal{D}_1$ $x$% Replay", where $x$ is the percentage of data in each training batch that comes from $\mathcal{D}_0$. Conversely, $(100\% - x)\%$ of the samples in each training batch will be sampled from $\mathcal{D}_1$. When comparing models trained with replay to other configurations, we ensure that the compute is *equivalent* by reducing the number of $\mathcal{D}_1$ tokens to accommodate replay tokens from $\mathcal{D}_0$.

## 5 Experimental Setup

To empirically evaluate the effectiveness of continually pre-training LLMs in comparison to training LLMs from a random initialization, we select recent pre-training datasets from the literature, outline practical continual pre-training settings for investigation, and select several baselines to compare with our proposed techniques. Our goal is to fairly compare our continual pre-training techniques to baselines in a controlled setting. We *do not* seek to obtain state-of-the-art performance or compare with models out of the scope of this paper.

### 5.1 Datasets

We use three datasets for training and validation: SlimPajama (Soboleva et al., 2023), German Common-Crawl (Laippala et al., 2022), and Pile (Gao et al., 2020). For all datasets, use the same tokenizer as Black et al. (2022) trained specifically on the Pile. To create our training set for SlimPajama, we randomly sub-sample the dataset (606B Total Tokens) to form a ∼299B token subset (see Table 1[1]) that is of comparable size to Pile. We also further sub-sample this SlimPajama subset to create three ∼ 100B token splits of the dataset (see Sec. 7.4 for details). For each of these datasets, we follow standard practice in LLM pre-training and select sampling percentages proportionally to the amount of data available in each domain such that one pass over the dataset does not repeat samples from any domain. To create the SlimPajama validation set we simply tokenize the default validation set that has been extensively deduplicated (Soboleva et al., 2023). To create the German training and validation sets, we split and tokenized the German Common Crawl scrape, available as part of the Oscar Dataset (Laippala et al., 2022), into a 195.43B token training set and a 982.6M token validation set. The Pile dataset comes pre-shuffled and mixed, we simply used the default training and validation sets. The training set is ∼ 330B tokens total, though in our experiments we only train on a 300B token subset.

Table 1: **Domain sizes of the 300B token training set of SlimPajama.** We sub-sampled the SlimPajama dataset (606B total tokens) into a 300B token split to make it of comparable size to Pile. We report the size of the subsampled domains that make up SlimPajama and the sampling percentage used at training time (e.g., the percentage of samples in each batch that come from a certain domain).

| Dataset | Size (Tokens) | Sampling (%) |
|---|---|---|
| Wikipedia | 11.96B | 4.00 |
| Book | 12.58B | 4.20 |
| C4 | 79.87B | 26.69 |
| Stack Exchange | 10.09B | 3.37 |
| GitHub | 15.63B | 5.22 |
| Common Crawl | 155.89B | 52.09 |
| Arxiv | 13.25B | 4.43 |
| Total | 299.28B | 100.00 |

### 5.2 Continual Learning Settings

We consider three realistic continual pre-training settings in the main body and provide results for a third which we believe is less warranted in the appendix. Each setting was carefully selected to expose different challenges and strengths of continual pre-training. Our setups assume that continually pre-trained LLMs undergo two or more pre-training phases sequentially. At the start of each phase, we reset the optimizer states, since optimizer states may not always be available, e.g. when using open-weight models from HuggingFace. That is, our results apply to situations where a continually pre-trained LLM is randomly initialized and pre-trained on datasets $\mathcal{D}_0, \mathcal{D}_1, \ldots, \mathcal{D}_{N-1}$ in sequence where $N \geq 2$. For the realistic settings we consider $tokens(\mathcal{D}_i) \geq 100B$. In each case, we consider the following natural baselines:

---

[1]We refer readers to the Pile paper (Gao et al., 2020) for its composition.

- A model trained from random initialization on the union of all datasets i.e. $\bigcup_{i=0}^{N-1} \mathcal{D}_i$, and

- A model trained from random initialization on individual dataset $\mathcal{D}_i$, $0 \leq i \leq N$.

$N = 2$ **settings** – Here we assume a model is available (e.g. via hugging face or pre-trained in-house) that has been pre-trained for autoregressive language modeling on a dataset ($\mathcal{D}_0$) using a linear warmup and cosine decay LR schedule. We also assume that the schedule follows existing conventions in the literature (e.g. decaying to the token budget; see Sec. 4 for details) as it is the case for most performant pre-trained LLMs (Rae et al., 2021; Hoffmann et al., 2022; Touvron et al., 2023a;b). Given a model pre-trained on $\mathcal{D}_0$, we now assume that a practitioner wants to update this model on a new dataset $\mathcal{D}_1$ using the same self-supervised objective. We consider the following concrete variations of the **two-dataset setting**:

- **Two datasets, weak shift**: In this variation, we consider $\mathcal{D}_0$ to be the Pile (Gao et al., 2020) and $\mathcal{D}_1$ to be pre-training on SlimPajama (Soboleva et al., 2023). SlimPajama is an extensively deduplicated version of RedPajama (Computer, 2023) which is built based on the LLaMA dataset (Touvron et al., 2023a). We consider this to be a weak but realistic distribution shift as both datasets are English-language and contain overlapping domains (CommonCrawl, GitHub, Arxiv, Wikipedia, StackExchange, Book, and C4), but SlimPajama (2023) is a newer dataset than Pile (2020) and is, therefore, likely to have newer data within these overlapping domains. Therefore, despite the potential for significant overlap, we believe this transition is realistic and is likely to be of interest to practitioners wishing to update an LLM on a similar distribution to pre-training (e.g., newly collected data of the same sources with higher quality filtering).

- **Two datasets, stronger shift**: In this variation, we consider $\mathcal{D}_0$ to be pre-training on the Pile (Gao et al., 2020) and $\mathcal{D}_1$ to be pre-training on German Common Crawl. German Common Crawl is a $\sim 200B$ token dataset taken from the Oscar dataset (Laippala et al., 2022). We note that this constitutes a stronger shift given the change of language. This setting is of particular interest for practitioners wishing to augment an LLM with a new natural language, programming language, or specific domain that is notably different in vocabulary from pre-training. We note, however, that as the domain strays farther and farther away from the tokenizer's training corpus, the tokenizer may become a key bottleneck to performance. We leave the treatment of the tokenizer to future work.

$N > 2$ **settings** – We also consider the following settings with more dataset transitions to investigate how well the methods considered scale with more datasets:

- **Three datasets, no shift** : We consider an $N = 3$ setting, where $\mathcal{D}_0, \mathcal{D}_1, \mathcal{D}_2$ are each district 100B token splits of SlimPajama. This setting is primarily used to evaluate the ability of our techniques to scale to many future updates and to assess the performance of our proposed infinite learning rate schedules.

- **Domain incremental continual pre-training**: This setting considers consuming the tokens of SlimPajama sequentially ordered by domain. That is, we train on a sequence of $N$ future datasets $\{\mathcal{D}_0, \mathcal{D}_1, \ldots, \mathcal{D}_{N-1}\}$ each of is a distinct domain of SlimPajama 300B. We note that this is similar to DACPT (Ke et al., 2022), however, we consider much larger datasets for each domain. This setting is particularly challenging due to the distribution shift experience at the transition between each domain. While it is certainly interesting, we believe it is unnecessarily difficult compare to mixing the SlimPajama data before training on it. The poor results in this setting (Sec. A.1 of the appendix) suggest that general-purpose LLMs should be continually pre-trained on a mixture of domains if possible, not updated per domain.

### 5.3 Training Setup

Using GPT-NeoX (Andonian et al., 2021) based on Megatron-DeepSpeed (Shoeybi et al., 2019; Microsoft, 2020), we train autoregressive decoder-only transformers with a causal language modeling objective. The

models use Pre-LN. Each model is trained using the same tokenizer as Black et al. (2022), which was trained exclusively on the Pile via the BPE algorithm (Sennrich et al., 2016). For all models, we train with the AdamW optimizer (Loshchilov & Hutter, 2019) using a batch size of 1104 and a sequence length of 2048. An epoch of training approximately corresponds to 132, 366 total training steps. As mentioned in the previous section, we reset the optimizer states between datasets. We consider two model sizes 405M and 9.6B parameters (referred to as 10B in this work) including embeddings. We train the smaller models using data parallelism across 46 6-GPU nodes using a micro-batch size of 4. The larger model is trained using tensor parallelism(Shoeybi et al., 2020) spanning six GPUs within a node and pipeline parallelism(Huang et al., 2019) spanning four nodes; that is, each model replica spans 24 GPUs across four nodes. We train this model on 276 nodes using gradient accumulation of 4 steps. Each model uses optimizer sharding via ZeRO-1 (Rajbhandari et al., 2020), activation checkpointing (Chen et al., 2016), activation partitioning across tensor parallel ranks, and mixed precision FP16/FP32 to reduce GPU memory consumption and fully utilize NVIDIA tensor cores during training. We provided an extended description of all hyperparameters in the appendix (Table. 13).

### 5.4 German and English LM Evaluation Benchmark

We measure performance on a wide variety of downstream tasks, which can be broadly categorized as follows.

English Benchmarks

- **Commonsense Reasoning (0-shot):** HellaSwag (Zellers et al., 2019), Winogrande (Sakaguchi et al., 2019), PIQA (Bisk et al., 2019), OpenBookQA (Mihaylov et al., 2018), ARC-Easy, ARC-Challenge (Clark et al., 2018)
- **World Knowledge (5-shot):** NaturalQuestions (Kwiatkowski et al., 2019), TriviaQA (Joshi et al., 2017)
- **Reading Comprehension (0-shot):** BoolQ (Clark et al., 2019)
- **Math:** MathQA (Amini et al., 2019)
- **Popular Aggregated Results:** MMLU (5-shot) (Hendrycks et al., 2021)

German Benchmarks from (Plüster, 2023), which translated their English counterparts using GPT 3.5 API

- **Commonsense Reasoning (0-shot):** HellaSwag-DE (Zellers et al., 2019), ARC-Challenge-DE (Clark et al., 2018)
- **World Knowledge (5-shot):** TriviaQA-DE (Joshi et al., 2017)
- **Popular Aggregated Results:** MMLU-DE (5-shot) (Hendrycks et al., 2021)

## 6 Results

We focus on continual pre-training when incoming datasets are large (200B tokens+). In such settings, training is expensive, thus, it is critical to efficiently adapt to the large amount of incoming data. However, most performant LLMs (Rae et al., 2021; Hoffmann et al., 2022; Zhao et al., 2023; Touvron et al., 2023b;a) are trained with a linear warmup and cosine decay schedule with a relatively low minimum learning rate. We hypothesize that **re-warming** this learning rate to a relatively high value and subsequently re-decaying it is needed to efficiently adapt to the new dataset. To this end, in section 6.1 we study the effect of linear warmup duration, re-warming the LR, re-decaying the LR, and maximum learning rate magnitude on adaptation and forgetting. Finding that re-warming and re-decaying increases both adaptation and forgetting, in section 6.2 we investigate whether replay can help mitigate forgetting when the learning rate is re-warmed and re-decayed. Subsections 6.3 and 6.4 combine the strategies studied in the previous two sections and report their performance relative to baselines for weak and strong distribution shifts and at large model scale. Finally, in section 7, we illustrate LR re-warming can cause unwanted forgetting, introduce infinite learning rate schedules as a promising way to circumvent it, and compare these schedules to baselines.

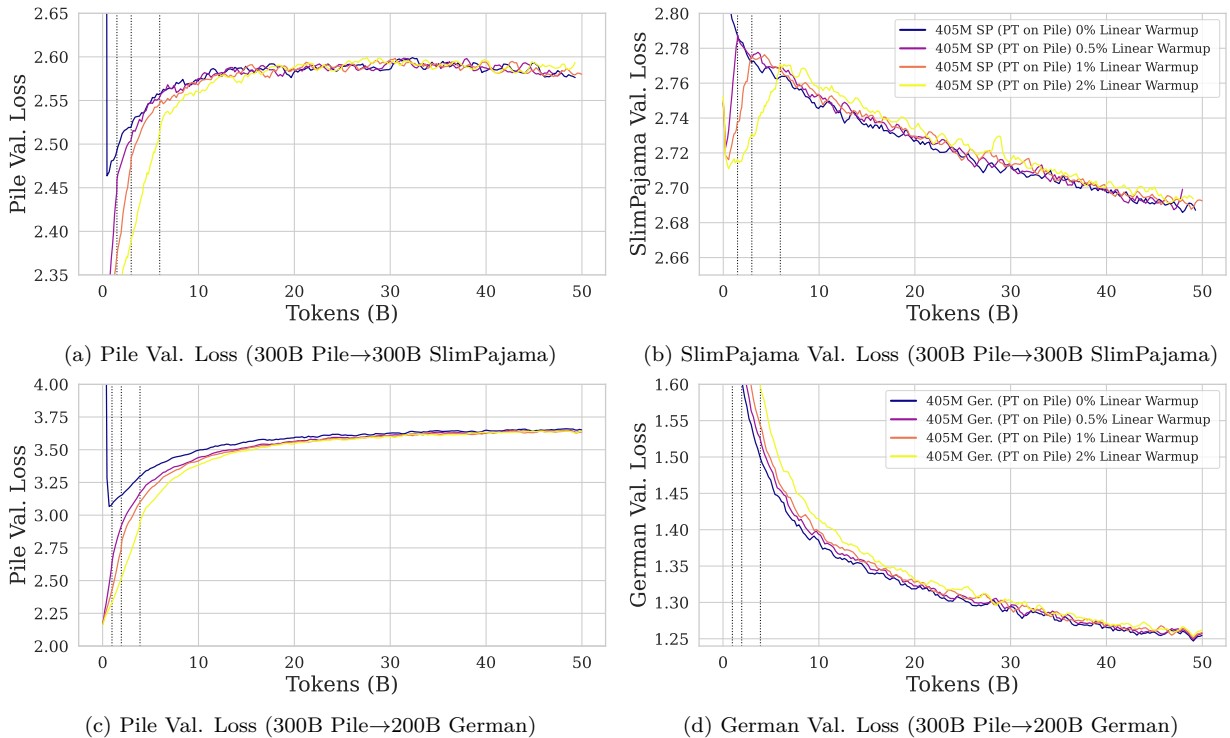

(a) Pile Val. Loss (300B Pile→300B SlimPajama)

(b) SlimPajama Val. Loss (300B Pile→300B SlimPajama)

(c) Pile Val. Loss (300B Pile→200B German)

(d) German Val. Loss (300B Pile→200B German)

Figure 3: **The effect of linear warmup for weak and strong distribution shifts.** (a),(b) and (c),(d) have the same legends respectively, shown in the right figures. We train 405M parameters models following a linear warmup and cosine decay schedule with varying linear warmup durations: 0%,0.5%,1%, and 2% of training iterations. Each learning rate schedule decays to $0.1\eta_{max}$ by the end of training based on the size of the dataset. We report results for the first 50B tokens of training. In the settings explored, we observe that the duration of the warm-up phase does not appear to be impactful when continuing to pre-train.

## 6.1 Learning Rate Schedule

Given the influence that the learning rate can have on adaptation and the low final LR values of prominent LLMs (Rae et al., 2021; Hoffmann et al., 2022; Zhao et al., 2023; Touvron et al., 2023b;a), we hypothesize that the LR should be re-warmed and re-decayed to promote adaptation during continual pre-training. In this section, we investigate the effect of linear warmup duration, re-warming the LR, re-decaying the LR, and the magnitude of the $\eta_{max}$ when continuing to pre-train. Specifically, we evaluate their respective effects in the **two-dataset weak shift** setting (300B Pile → 300B SlimPajama) and the **two-dataset stronger shift** setting (300B Pile → 300B SlimPajama). Notably, the model trained on $\mathcal{D}_0$ (300B tokens of Pile) follow a linear warmup and cosine decay schedule[2], simulating many common open-source pre-trained LLMs.

### 6.1.1 The Effect of Linear Warmup for Weak and Strong Distribution Shifts.

We first investigate the effect of linear warm-up duration on forgetting and adaptation in the **two datasets, weak shift** and **two datasets, stronger shift** settings (see Sec. 5.2 for details). The models are pre-trained on 300B tokens of Pile (Gao et al., 2020) ($\mathcal{D}_0$). We continue to pre-train the models on SlimPajama (weak shift) and German Common Crawl (stronger shift) for the first 50B tokens of training. We re-warm and re-decay the learning rate using a cosine learning rate schedule set to reach its minimal value ($\eta_{min} = 0.1 \cdot \eta_{max}$) at 300B and 200B tokens, respectively. We consider warming up the learning rate for 0.5%, 1%, and 2% of $\mathcal{D}_1$'s total training iterations (132366 and 86000 iterations, respectively). Since the decay happens over the

---

[2]For all cosine decays in this paper, unless otherwise specified, we fit the cosine annealing phase to the token budget, set the linear warmup duration ($T_{warmup}$) to 1% of training iterations, and set $\eta_{min} = 0.1 \cdot \eta_{max}$

remaining budget of iterations (so resp. $99.5\%, 99\%$ and $98\%$ of the total iterations), note that this implies that the decay phase of longer warmups happens marginally faster. Additionally, we train a model with no linear warm-up ($0\%$) that immediately decays the LR from $\eta_{max}$. All experiments are conducted on a 405M parameter model.

Figure 3 reports the validation losses for $\mathcal{D}_0$ and $\mathcal{D}_1$ for all models throughout the first 50B tokens of continued pre-training on $\mathcal{D}_1$. The top row reports results for the weak distribution shift, while the bottom row reports results for the stronger distribution shift. Across both distribution shifts, we observe that models using shorter linear warmup initially forget and adapt faster than their longer warmup counterparts. This happens because they increase the LR faster which leads to faster forgetting and adaptation. In particular, the model without any warmup adapts and forgets the fastest—even undergoing an initial chaotic phase (as seen in the continual learning literature (De Lange et al., 2022)). Indeed, coupled with noisy gradients due to adapting to a new distribution and the resetting of optimizer states, its large initial learning rate causes a transient spike in validation loss across both shifts. In all scenarios, however, these initial differences diminish throughout training, leaving all models with relatively similar forgetting and adaptation after 50B tokens.

*Thus, in the settings explored, the duration of the linear warm-up phase does not appear to affect forgetting or adaptation as measured by the validation loss when continuing to pre-train, although it can prevent initial transient spikes in the loss.*

With this in mind, we set a linear warmup duration of $1\%$ of training iterations for all subsequent experiments.

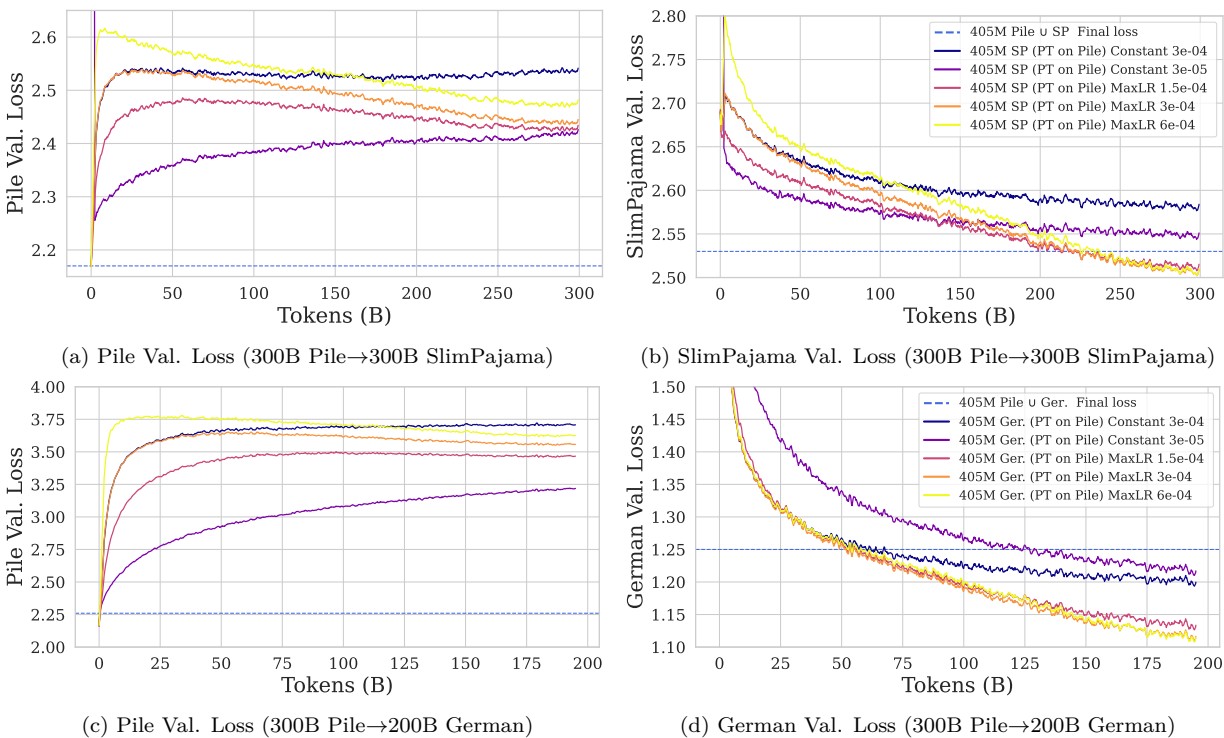

(a) Pile Val. Loss (300B Pile→300B SlimPajama)

(b) SlimPajama Val. Loss (300B Pile→300B SlimPajama)

(c) Pile Val. Loss (300B Pile→200B German)

(d) German Val. Loss (300B Pile→200B German)

Figure 4: **The effect of re-warming and re-decaying the learning rate on adaptation and forgetting.** We consider two constant baselines and three models that re-warm and re-decay. One baseline continues training from $\eta_{min}$ of pre-training ($3 \cdot 10^{-4}$) while the other warms up to $\eta_{max}$ from pre-training ($3 \cdot 10^{-4}$). For the models that re-warm and re-decay we vary $\eta_{max} \in \{1.5 \cdot 10^{-4}, 3 \cdot 10^{-4}, 6 \cdot 10^{-4}\}$. All models except the $\eta_{min}$ baseline use linear warmup for $1\%$ training iteration. The non-baseline models cosine decay the learning to reach $0.1 \cdot \eta_{max}$ by the end of training. We observe that re-warming and re-decaying the learning rate is needed to best adapt to the new dataset. Small increases or decreases in $\eta_{max}$ allow to trade-off between more or less adaptation. A stronger distribution shift seems to be a catalyst for both forgetting and adaptation.

### 6.1.2 The effect of re-warming, re-decaying, and varying $\eta_{max}$ for Weak and Strong Distribution Shifts.

We now investigate the benefits of re-warming and re-decaying the learning rate (e.g., following a cosine schedule) for different values of $\eta_{max}$. Specifically, we compare these models to two natural baselines: a model that does not re-warm, staying constant at $\eta_{min}$ ($3 \cdot 10^{-5}$), and a model that re-warms to the pre-training $\eta_{max}$ ($3 \cdot 10^{-4}$) but does not re-decay. We use the same two two-dataset settings: we first pre-train on the Pile ($\mathcal{D}_0$) for 300B tokens and continually pre-train our model on SlimPajama (weak shift) or German Common Crawl (strong shift) as our $\mathcal{D}_1$ datasets. The continual pre-training is conducted for the full size (300B and 200B tokens, respectively) of the datasets. The models that re-warm and re-decay the LR consider three strategies: re-warming to half the pre-training's $\eta_{max}$ ($1.5 \cdot 10^{-4}$), re-warming to the same $\eta_{max}$ as pre-training ($3 \cdot 10^{-4}$), and re-warming to twice the $\eta_{max}$ of pre-training ($6 \cdot 10^{-4}$). In all cases, the learning rate is cosine-decayed after linear warmup to reach $\eta_{min} = 0.1 \cdot \eta_{max}$ by the end of training. Finally, we consider models trained on $\mathcal{D}_0 \cup \mathcal{D}_1$ as a third baseline (union-trained) to provide an upper bound on performance.

Figure 4 reports validation losses for the $\mathcal{D}_0$ and $\mathcal{D}_1$ datasets throughout the continual pre-training of all models. The top row of plots reports results for the weak distribution shift (300B Pile→300B SP), while the bottom row reports results for the stronger distribution shift (300B Pile→200B Ger.). For both shifts, the constant $\eta_{min}$ learning rate model achieves the least forgetting on $\mathcal{D}_0$. It also adapts the least on $\mathcal{D}_1$ for the stronger shift, however, for the weak shift it adapts more than the constant $\eta_{max}$ baseline. When comparing these constant LR baselines to the models that re-warm and re-decay on both shifts considered, we observe that the latter models adapt better to the new dataset by a significant margin for both distribution shifts. This shows that re-warming and re-decaying are necessary to maximize adaptation to the new dataset when continually pre-training LLMs. Among the models that re-warm and re-decay the LR, we observe that varying the learning rate causes small differences in adaptation and forgetting: higher values of $\eta_{max}$ lead to more forgetting and more adaptation while the opposite is true for lower values. When comparing the constant LR baselines to the union-trained baseline, we observe that the final validation loss for $\mathcal{D}_0$ is significantly higher than the union-trained model's on both distribution shifts. This is also the case for $\mathcal{D}_1$ on the weak distribution shift, but interestingly for the stronger distribution shift, the constant baselines achieve lower $\mathcal{D}_1$ validation loss than the union-trained model. The stronger distribution shift appears to exacerbate the relative forgetting and ability of the models to adapt in the context of continually pretrained LLMs. When comparing models continually pre-trained with re-warming and re-decaying to the union baseline, we note that these models adapt better (lower final validation loss) to $\mathcal{D}_1$ than the union baseline. However, these models experience significant forgetting on $\mathcal{D}_0$, showing the need for replay to make these models competitive with the union baseline.

*In summary, continually pre-training LLMs, both re-warming and re-decaying are necessary to maximize adaptation to the new dataset; small increases or decreases in $\eta_{max}$ allow to trade-off between more or less adaptation; a stronger distribution shift between $\mathcal{D}_0$ and $\mathcal{D}_1$ exacerbates forgetting and enhances adaptation; and the duration of linear warm-up phase does not appear to be impactful on forgetting or adaptation.*

## 6.2 The Effect of Replay

In this subsection, we explore the effect of compute-equivalent replay when continually pre-training models that re-warm and re-decay the learning rate.

Given the need to mitigate forgetting when re-warming and re-decaying, we move on to investigate the effects of replay in our weak and strong-shift continued pre-training scenarios. Specifically, we use compete equivalent replay (see Sec. 4.2 for details) where replay tokens from $\mathcal{D}_0$ are added at the cost of removing the equivalent number of $\mathcal{D}_1$ tokens from the budget. Following the same two dataset settings, the model is pre-trained on $\mathcal{D}_0$ (Pile) for 300B tokens. This is followed by continual pre-training on a SlimPajama (weak shift) or German Common Crawl (strong shift). For more details regarding the setup, please see Section 5.2. Our continued pre-training is conducted for the full size of the respective datasets, which is 300B tokens for SlimPajama (weak shift) and 200B tokens for German Common Crawl (strong shift). We consider 1%, 5%, 10%, and 50% replay for both shifts and add 0.5% and 25% replay runs for the weak and strong distribution shifts respectively. We consider two baselines to put these results into a broader context. The first baseline is a model trained on $\mathcal{D}_1$ without replay. The second baseline model is trained from random initialization on a

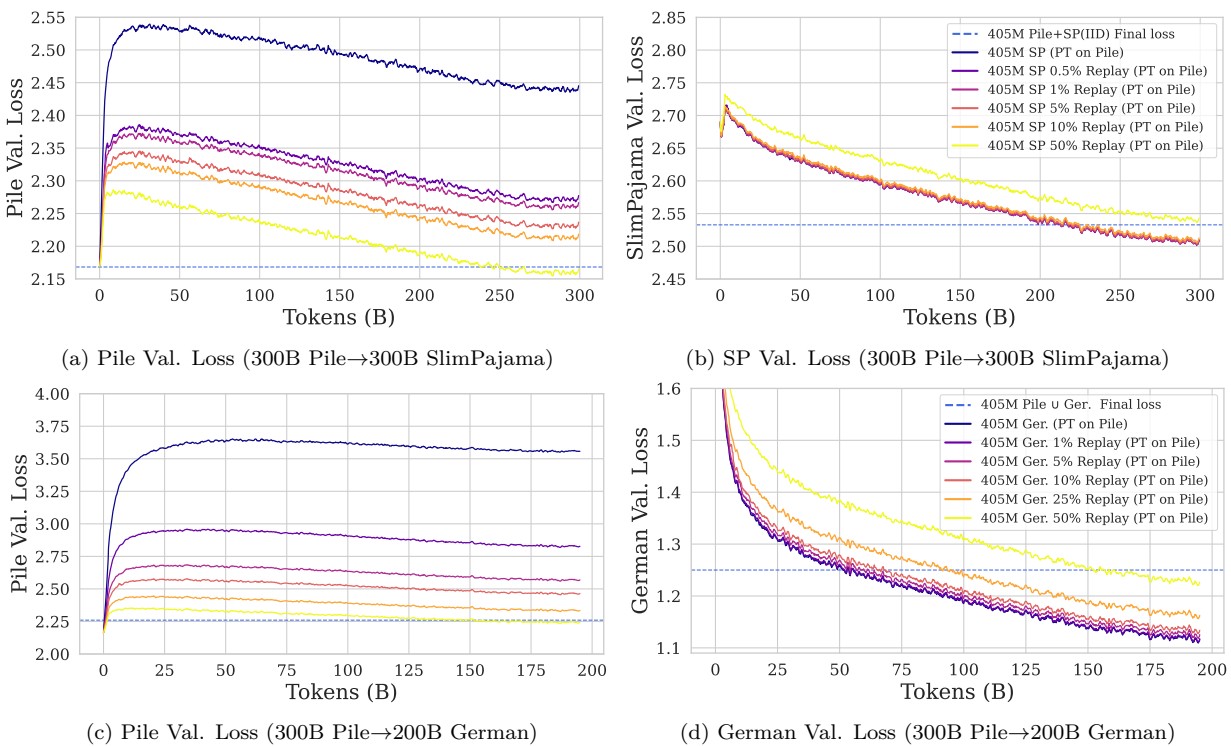

Figure 5: **The effect of replay at 405M scale for weak and strong distribution shifts.** We report Pile validation loss (left) and SlimPajama/German validation (right top/bottom) during training. Each model is trained from a checkpoint pre-trained on 300B tokens of Pile. The blue dotted line reports the final validation loss for models trained on Pile∪SlimPajama or Pile∪German data, totaling 600B and 500B tokens datasets respectively. We observe that replay significantly reduces forgetting across both shifts, however, the stronger shift requires more replay to mitigate forgetting to the same extent.

union of $\mathcal{D}_0$ and $\mathcal{D}_1$ for 600B tokens (SlimPajama) and 500B tokens (German Common Crawl). The latter baseline reflects the practice of fully re-training the model to update it instead of continually pre-training the existing model. All models re-warm and re-decay the learning rate using a cosine decay schedule fit to their token budget with the same $\eta_{max}$ ($3 \cdot 10^{-4}$) and $\eta_{min}$ ($3 \cdot 10^{-5}$) values as during pre-training on $\mathcal{D}_0$.

**Validation Loss Comparison** The results in Fig. 5 (top and bottom) show the evolution of the validation loss during continual pre-training on the respective $\mathcal{D}_1$ datasets. Table 2 reports the average final validation loss for each of these models. The final loss is averaged over the last 100 iterations of training sampled at intervals of 10 iterations. We consistently observe across both distribution shifts that even the lowest tested replay of 1% significantly reduces forgetting on Pile compared to the no-replay baselines. This effect is more pronounced in the strong-shift scenario due to the larger amount of forgetting in this setting. We observe little impact on downstream performance for 1%, 5%, and 10% replay when compared to the 0% baseline, showing that the forgetting benefits of replay come at little cost in our setting. However, when using an extreme amount of replay (50%), we observe that the model adapts relatively significantly worse to $\mathcal{D}_1$. Interestingly, for both datasets, the 50% replay models attain or surpass the final average validation performance of the baseline training on $\mathcal{D}_1 \cup \mathcal{D}_0$. This is curious as these model have seen 150B (for SlimPajama) and 100B (for German) fewer tokens of $\mathcal{D}_1$ than their respective baselines.

_In summary, we find that, when re-warming and re-decaying the LR in a continual pre-training context, replay is a useful tool for reducing forgetting. For both distribution shifts, using an appropriate amount of replay yields similar final validation loss to the $\mathcal{D}_1 \cup \mathcal{D}_0$ baseline. Moreover, for both shifts, the use of replay_

Table 2: **Final loss of English-only 405M parameter models trained with varying amounts of replay.** The loss is averaged over the last 100 iterations of training sampled at intervals of 10 iterations. The standard error for these measurements was computed but is not reported as it was $< 0.001$ for all models. We observe that models using more replay achieve a better adaptation-forgetting trade-off (AVG Loss). Interestingly, the model using 50% replay archives nearly identical loss values while seeing 150B fewer tokens on SlimPajama.

| Training Tokens | Validation Loss | | |
|---|---|---|---|
| | $\mathcal{D}_0$ Pile | $\mathcal{D}_1$ SlimPajama/German | AVG |
| 300B Pile $\rightarrow$ 300B SP | 2.44 | 2.50 | 2.47 |
| 300B Pile $\rightarrow$ 300B SP (0.5% Replay) | 2.27 | 2.50 | 2.39 |
| 300B Pile $\rightarrow$ 300B SP (1% Replay) | 2.26 | 2.50 | 2.38 |
| 300B Pile $\rightarrow$ 300B SP (5% Replay) | 2.23 | 2.51 | 2.37 |
| 300B Pile $\rightarrow$ 300B SP (10% Replay) | 2.21 | 2.51 | 2.36 |
| 300B Pile $\rightarrow$ 300B SP (50% Replay) | 2.16 | 2.54 | **2.35** |
| 600B Pile $\cup$ SP | 2.17 | 2.53 | **2.35** |
| 300B Pile $\rightarrow$ 200B Ger. | 3.56 | 1.11 | 2.34 |
| 300B Pile $\rightarrow$ 200B Ger. (1% Replay) | 2.83 | 1.12 | 1.97 |
| 300B Pile $\rightarrow$ 200B Ger. (5% Replay) | 2.57 | 1.12 | 1.85 |
| 300B Pile $\rightarrow$ 200B Ger. (10% Replay) | 2.46 | 1.13 | 1.80 |
| 300B Pile $\rightarrow$ 200B Ger. (25% Replay) | 2.33 | 1.16 | 1.75 |
| 300B Pile $\rightarrow$ 200B Ger. (50% Replay) | 2.24 | 1.22 | **1.73** |
| 500B Pile $\cup$ Ger. | 2.26 | 1.25 | 1.75 |

*seems to negligibly affect adaptation to the downstream dataset, showing that reducing forgetting via replay comes at very little cost when continually pre-training LLMs.*

### 6.3 Continual Pre-training Final Performance for Weak and Strong Distribution Shifts.

In this subsection, we compare two continually pre-trained 405M parameter models to several baselines in the *two dataset weak shift* (Pile $\rightarrow$ SlimPajama) and *two dataset strong shift* (Pile $\rightarrow$ German) settings. Our main goal is to determine how the differences in distribution shift affect final performance.

**Continually Pre-trained Models** To ablate the performance of combining LR re-warming and re-decaying with replay, we opt to train one model that exclusively re-warms and re-decays the learning rate and another that combines both techniques. Given results from the previous section showing that many replay percentages obtain similar average validation loss, we select 5% replay for the weak shift setting and 25% replay for the stronger shift setting because these percentages allow us to see more new tokens than their higher replay counterparts (due to compute-equivalent replay) with a similar average final validation loss. For both models, we re-warm to the $\eta_{max}$ of pre-training ($3 \cdot 10^{-4}$) and re-decay it using a cosine decay schedule set to reach $\eta_{min}$ by the end of continual pre-training. More hyperparameters are reported in Table 13 of the appendix.

**Baselines** We also train several baselines. Two baselines are trained on $\mathcal{D}_0$ and $\mathcal{D}_1$ respectively while the third is trained on the union of each dataset $\mathcal{D}_0 \cup \mathcal{D}_1$. We consider the model trained on $\mathcal{D}_0 \cup \mathcal{D}_1$ to be an upper bound on performance as it represents an expensive full re-training. The baselines trained on individual datasets can be seen as compute-equivalent alternatives to continual pre-training (e.g., one could opt to train a model from random initialization on $\mathcal{D}_1$ instead of continually pre-training it).

#### 6.3.1 Final Performance Evaluated by Loss

Figure 6 reports the validation loss during continual pre-training of 405M parameter models for weak (top) and strong (bottom) shifts. Table 3 reports the average (over the last 100 iterations) final loss value for these models. Since the transition from English to German represents a starker distribution shift than Pile to SlimPajama, training on German leads to significantly more forgetting on Pile ($\mathcal{D}_0$) for the continually pre-trained model without replay (0.27 vs 1.39 for weak and strong shifts respectively). However, choosing 25% replay to handle the starker shift significantly reduces the amount of forgetting on Pile, a reduction of 1.23 in terms of final loss. When comparing continually pre-trained models to baselines trained exclusively

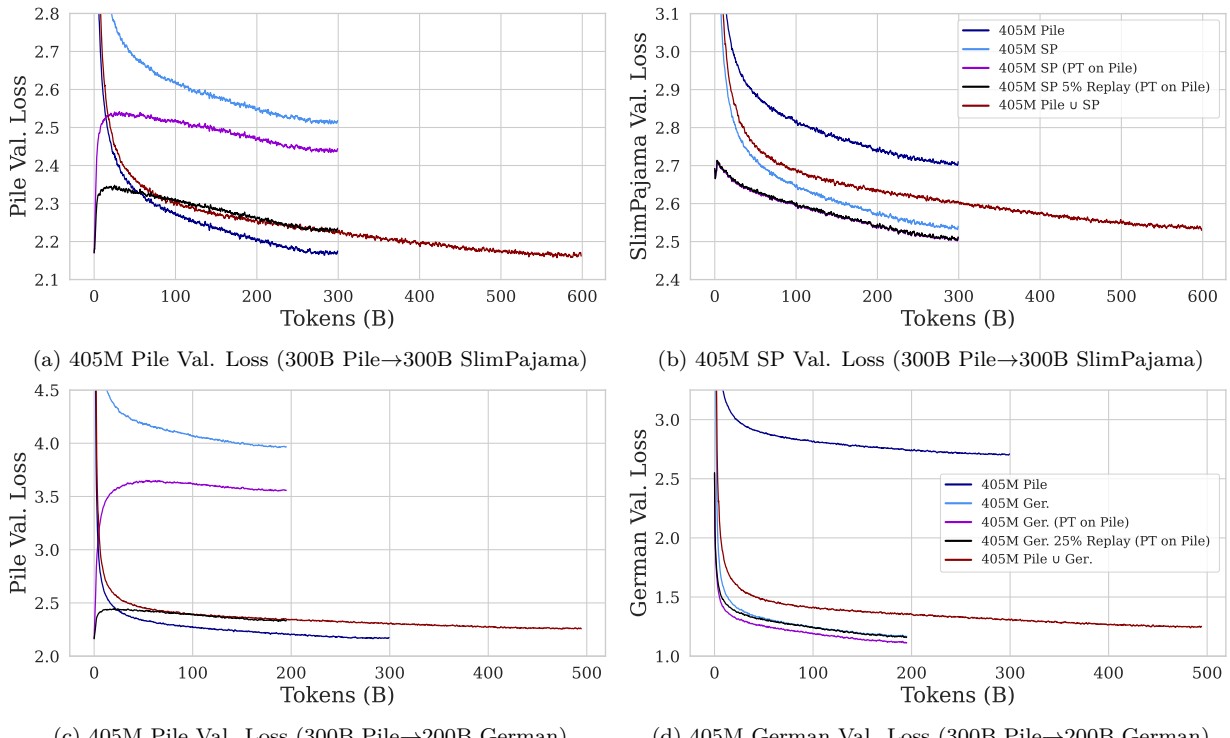

(a) 405M Pile Val. Loss (300B Pile→300B SlimPajama)

(b) 405M SP Val. Loss (300B Pile→300B SlimPajama)

(c) 405M Pile Val. Loss (300B Pile→200B German)

(d) 405M German Val. Loss (300B Pile→200B German)

Figure 6: **Final loss of 405M parameter models trained on two distribution shifts.** Figures (a) and (b) are duplicated from Fig. 7 for convenient comparison. we provided three baselines and two continually pre-trained models. The baselines (light blue, dark blue, and maroon) are trained from random initialization on 300B tokens of SlimPajama, 300B tokens of Pile, and the union of both datasets (600B tokens). The continually pre-trained models (black and violet) start from a checkpoint pre-trained on 300B tokens of Pile (dark blue curve) and use 0% and 5% replay, respectively. We observe that for both distribution shifts, the combination of re-warming the learning rate and using a small percentage of replay helps to strike a balance between forgetting and adaptation. Importantly, we note that the use of replay minimally affects downstream performance compared to the models using 0% replay.

Table 3: **Final loss of continually pre-trained English-only & English-German models.** All models have 405M parameters. The loss is averaged over the last 100 iterations of training sampled at intervals of 10 iterations. The standard error for these measurements was computed but is not reported as it was $< 0.001$ for all models. We observe that even for starker distribution shifts, the combination of LR warmup and 25% replay helps to match the average performance of the Pile $\cup$ German model.

| Training Tokens | Validation Loss | | | LM Eval. Acc. | |
|---|---|---|---|---|---|
| | $\mathcal{D}_0$ **Pile** | $\mathcal{D}_1$ **German/SP** | **AVG** | **English** | **HellaSwag-DE** |
| 300B Pile | 2.17 | 2.70 | 2.44 | 33.95 | 27.09 |
| 300B SP | 2.51 | 2.53 | 2.52 | 34.11 | 27.03 |
| 300B Pile → 300B SP | 2.44 | 2.50 | 2.47 | 34.93 | 27.43 |
| 300B Pile → 300B SP (5% Replay) | 2.23 | 2.51 | **2.37** | 35.14 | 27.09 |
| 600B Pile $\cup$ SP | 2.17 | 2.53 | **2.35** | 34.30 | 27.36 |
| 300B Pile | 2.17 | 2.70 | 2.44 | 33.95 | 27.09 |
| 200B German | 3.97 | 1.17 | 2.57 | 27.74 | 29.53 |
| 300B Pile → 200B German | 3.56 | 1.11 | 2.34 | 29.20 | 31.23 |
| 300B Pile → 200B German (25% Replay) | 2.33 | 1.16 | **1.75** | 32.48 | 31.04 |
| 500B Pile $\cup$ German | 2.26 | 1.25 | **1.75** | 32.43 | 30.45 |

on $\mathcal{D}_1$, we observe that the continually pre-trained models always have lower validation loss across both distribution shifts. When comparing the continually pre-trained models with the $\mathcal{D}_0 \cup \mathcal{D}_1$ baselines we find that both models achieve nearly identical (weak shift) or identical (strong shift) average final validation losses.

This shows that for strong and weak distribution shifts, a simple and scalable combination of LR re-warming, LR re-decaying, and replay can achieve similar performance to the $\mathcal{D}_0 \cup \mathcal{D}_1$ baseline.

### 6.3.2 Final Performance Evaluated by Zero-shot and Few-shot Results on Popular LM Benchmarks

While final accuracy provides a good measure of performance on the pre-training objective, LLMs' abilities are typically judged by their performance on evaluation tasks. With the caveat that we use base models, that is our models have not been instruction-tuned, fine-tuned, or adapted to human preferences in any way, we present their evaluation on popular benchmarks in this section. Furthermore, we also provide a qualitative evaluation of German-trained models. We refer the reader to Sec. 5.4 of the main manuscript and Sec. A.6 of the appendix for a more detailed description of the chosen evaluation tasks.

Table 3 reports the average accuracy of each model for our English evaluation tasks and the normalized accuracy for the German HellaSwag evaluation task. We do not report the average German evaluation score as it is not informative due to evaluations having near-random chance accuracy (see Table 11). We observe that English models consistently outperform German models on the English evaluations. However, the strong replay used with the 25% replay German model helps to reduce this gap. English models' English evaluation performance is very similar with a range of 1.19 between the highest and lowest values. We suspect that there is significant noise in the evaluation process for base models of this size and believe that the differences are likely not significant. That being said, the continually pre-trained model with LR re-warming, LR re-decaying, and replay does improve on the $\mathcal{D}_0 \cup \mathcal{D}_1$ model. When evaluating German-trained models on English evaluation tasks, we see consistent improvements for models using more replay. We note that once again the model trained with LR re-warming, LR re-decaying, and replay does improve on the $\mathcal{D}_0 \cup \mathcal{D}_1$ model. Turning to the German HellaSwag results we observe that German models consistently outperform their English counterparts. Among German-trained models, the continually trained models outperform the union-trained model and the model trained exclusively on German.

Given the poor performance of German models on all German evaluation tasks except HellaSwag (the same as English models on average), we further investigated their understanding of German by conducting a short qualitative study of model generations. In section A.5 of the appendix, we select five German prompts that contain various peculiarities of the German language (see Tab. 8 of the appendix). We then generate a fixed token-length response for each of the models trained German Common Crawl. As a baseline, we also evaluate the model trained only on the Pile. Despite the poor quality of generations at small model scale, we find that there is an observable improvement in the generative quality of German-language outputs from the models trained on German Common Crawl when compared to the Pile baseline, which tends to be systematically off-topic. This suggests that while our German-trained models have learned about the language, the evaluation tasks are too difficult to pick it up at the 405M parameter scale. Another reason is that the German dataset is smaller than the English datasets considered, and contains only web-scraped data, as opposed to the more sophisticated English datasets used in this work.

*In summary, for weak and stronger distribution shifts alike, it is possible to achieve competitive performance to a model trained on $\mathcal{D}_0 \cup \mathcal{D}_1$ by utilizing a simple and scalable combination of LR re-warming, LR re-decaying, and replay. This is true for final validation loss and averaged language model evaluation scores, showing that this powerful combination of simple techniques can equip language models with new knowledge with little compromise to existing knowledge.*

### 6.4 Continual Pre-training Final Performance at Different Model Scales

In this subsection, we establish the effect of increasing parameter count by an order of magnitude on the final performance of continual pre-training. To accomplish this we compare two continually pre-trained models to several baselines at 405M and 10B parameter model sizes in the *two dataset weak shift* (Pile $\rightarrow$ SlimPajama) and *two dataset strong shift* (Pile $\rightarrow$ German) settings.

**Continually Pre-trained Models**   To ablate the performance of combining LR re-warming and re-decaying with replay, we opt to train one model that exclusively re-warms and re-decays the learning rate and another that combines both techniques. Given results from (Sec. 6.2) for the weak distribution shifts, showing that

many replay percentages obtain similar average validation loss, we select 5% replay for both model scales because these percentages allow us to see more new tokens than their higher replay counterparts (due to compute-equivalent replay) with a similar average final validation loss. For both models, we re-warm to the $\eta_{max}$ of pre-training $(3 \cdot 10^{-4})$ and re-decay using cosine annealing set to reach $\eta_{min}$ by the end of continual pre-training. More hyperparameters are reported in Table 13 of the appendix.

**Baselines** We also train several baselines. Two baselines are trained on $\mathcal{D}_0$ and $\mathcal{D}_1$ respectively while the third is trained on $\mathcal{D}_0 \cup \mathcal{D}_1$. We consider the model trained on $\mathcal{D}_0 \cup \mathcal{D}_1$ to be an upper bound on performance as it represents an expensive full re-training. The baselines trained on individual datasets can be seen as compute-equivalent alternatives to continual pre-training (e.g., one could opt to train a model from random initialization on $\mathcal{D}_1$ instead of continually pre-training it).

### 6.4.1 Final Performance Evaluated by Loss

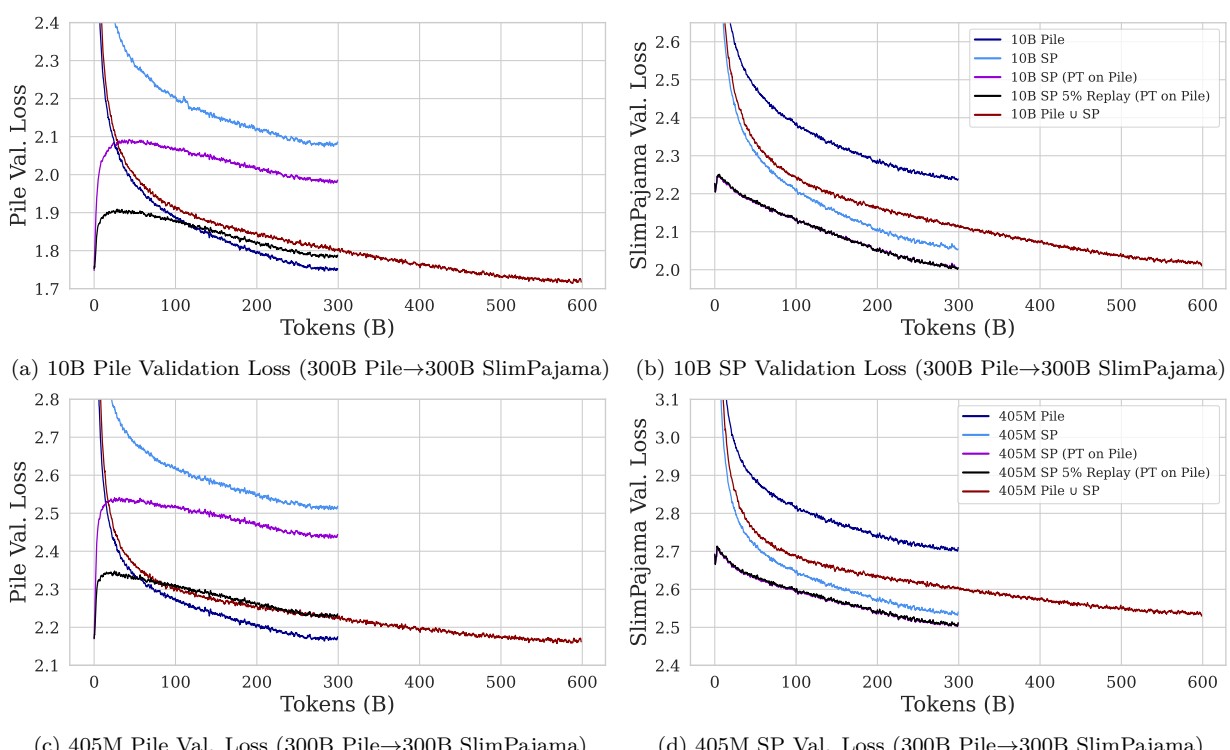

Figure 7: **Validation loss during continual pre-training of 10B (top) and 405M (bottom) parameter models.** At each model scale we provided three baselines and two continually pre-trained models. The baselines (light blue, dark blue, and maroon) are trained from random initialization on 300B tokens of SlimPajama, 300B tokens of Pile, and the union of both datasets (600B tokens). The continually pre-trained models (black and violet) start from a checkpoint pre-trained on 300B tokens of Pile (dark blue curve) and use 0% and 5% replay, respectively. We observe that for both model sizes, the combination of LR re-warming, LR re-decaying, and using a small percentage of replay helps to strike a balance between forgetting and adaptation. Importantly, we note that the use of replay minimally affects downstream performance compared to the models using 0% replay (black and violet curves overlap in figures (b) and (d)).

Figure 7 reports the validation loss during continual pre-training for 405M and 10B models, while Table 4 reports the average (over the last 100 iterations) final loss value for each model. As expected, we observe that all baselines and continually pre-trained models consistently improve in perplexity on both datasets from increasing parameter count. For the 405M models, we observe that Pile ∪ SP achieves identical validation loss on each dataset to the baselines trained individually on them. In contrast, the 10B parameter model

Table 4: **Final loss of 10B and 405M parameter models.** The loss is averaged over the last 100 iterations of training sampled at intervals of 10 iterations. The standard error for these measurements was computed but is not reported as it was $< 0.001$ for all models. We observe that at both model scales, learning rate re-warming combined with 5% replay approaches the average loss value of joint training.

| Model Size | Training Tokens | Validation Loss | | |
| | | $\mathcal{D}_0$**Pile** | $\mathcal{D}_1$**SlimPajama** | **AVG** |
| --- | --- | --- | --- | --- |
| 10B | 300B Pile | 1.75 | 2.24 | 1.99 |
| | 300B SP | 2.08 | 2.05 | 2.07 |
| | 300B Pile $\rightarrow$ 300B SP | 1.98 | 2.00 | 1.99 |
| | 300B Pile $\rightarrow$ 300B SP (5% Replay) | 1.79 | 2.00 | **1.89** |
| | 600B Pile $\cup$ SP | 1.72 | 2.02 | **1.87** |
| 405M | 300B Pile | 2.17 | 2.70 | 2.44 |
| | 300B SP | 2.51 | 2.53 | 2.52 |
| | 300B Pile $\rightarrow$ 300B SP | 2.44 | 2.50 | 2.47 |
| | 300B Pile $\rightarrow$ 300B SP (5% Replay) | 2.23 | 2.51 | **2.37** |
| | 600B Pile $\cup$ SP | 2.17 | 2.53 | **2.35** |

trained on Pile $\cup$ SP outperforms the models trained individually on each. We hypothesize that this happens due to larger models having more capacity, thus being capable of learning at a higher rate for longer. We observe that replaying 5% pile data when continuing to pre-train on SlimPajama reduces forgetting on Pile validation by 0.19 and 0.21 for 10B and 405M parameter models respectively. The negligible difference in forgetting-reduction from replay despite the order of magnitude difference in parameters between both models suggests that model scale has a limited negative influence on forgetting-reduction from replay. We believe this is because larger models forget less by default. Indeed, the models trained without replay from a pre-trained Pile checkpoint forget 0.23 and 0.27 nats of Pile perplexity for 10B and 405M respectively. While the difference is small, this suggests that larger models forget less, confirming our hypothesis. When comparing the average final validation loss of the models with 5% replay and baselines trained on the union of both datasets, we notice that there is only a difference of 0.02 for both model sizes. This shows that for weak but realistic distribution shifts at two model scales, continual pre-training can achieve similar performance to the expensive re-training baseline.

### 6.4.2 Final Performance Evaluated by Zero-shot and Few-shot Results on Popular LM Benchmarks

While final accuracy provides a good measure of performance on the pre-training objective, LLMs abilities are typically judged by their performance on evaluation tasks. With the caveat that we use base models, that is our models have not been instruction-tuned, fine-tuned, or adapted to human preferences in any way, we present their evaluation on popular benchmarks in this section. We refer the reader to Sec. 5.4 of the main manuscript and Sec. A.6 of the appendix for a more detailed description of the chosen evaluation tasks.

Table 5: **All Zero-shot and Few-shot results on popular LM benchmarks.** Normalized accuracy is reported for HellaSwag and exact match (EM) is reported for NaturalQuestions and TriviaQA. All other tasks report unnormalized accuracy. MMLU and TriviaQA are evaluated 5-shot, while all other tasks are zero-shot. We observe **on average**, as expected, that 10B parameter models outperform their 405M counterparts and that the English-only 405M models outperform their German-trained counterparts.

| Model Size | Training Tokens | HellaSwag | ARC-c | ARC-e | BoolQ | MathQA | MMLU | OBQA | PIQA | WG | TfQA1 | TfQA2 | NQ | TrQA | AVG |
| --- | --- | --- | --- | --- | --- | --- | --- | --- | --- | --- | --- | --- | --- | --- | --- |
| 10B | 300B Pile | 68.46 | 34.81 | 69.49 | 68.20 | 27.34 | 27.28 | 27.20 | 76.82 | 62.51 | 20.44 | 33.68 | 6.65 | 41.92 | 43.45 |
| | 300B SP | 70.38 | 36.77 | 71.93 | 68.04 | 24.76 | 27.42 | 28.20 | 76.99 | 65.04 | 22.40 | 33.99 | 11.25 | 52.63 | 45.37 |
| | 300B Pile $\rightarrow$ 300B SP | 73.66 | 37.37 | 73.02 | 73.18 | 26.43 | 29.94 | 30.20 | 78.51 | 66.30 | 23.26 | 35.04 | 12.99 | 57.94 | 47.53 |
| | 300B Pile $\rightarrow$ 300B SP (5% Replay) | 73.24 | 39.42 | 74.24 | 70.80 | 26.83 | 28.79 | 30.60 | 78.02 | 68.67 | 23.01 | 35.02 | 13.32 | 57.86 | 47.68 |
| | 600B Pile $\cup$ SP | 73.39 | 39.25 | 73.57 | 72.05 | 26.83 | 37.78 | 27.80 | 77.58 | 67.32 | 23.13 | 36.16 | 12.41 | 56.73 | 48.00 |
| 405M | 300B Pile | 40.95 | 22.01 | 51.77 | 59.24 | 24.12 | 26.18 | 19.80 | 66.59 | 53.83 | 24.85 | 42.11 | 0.91 | 8.97 | 33.95 |
| | 300B SP | 44.22 | 21.76 | 54.08 | 59.63 | 22.71 | 26.18 | 19.60 | 68.23 | 49.80 | 22.64 | 38.63 | 1.69 | 14.18 | 34.11 |
| | 300B Pile $\rightarrow$ 300B SP | 46.22 | 22.70 | 54.04 | 57.43 | 24.22 | 25.28 | 21.20 | 69.26 | 54.46 | 23.13 | 38.91 | 2.02 | 15.23 | 34.93 |
| | 300B Pile $\rightarrow$ 300B SP (5% Replay) | 46.55 | 23.55 | 55.01 | 57.92 | 24.22 | 25.94 | 20.60 | 69.37 | 54.22 | 23.38 | 38.35 | 1.99 | 15.70 | 35.14 |
| | 600B Pile $\cup$ SP | 45.06 | 23.55 | 52.99 | 55.57 | 23.12 | 26.65 | 18.20 | 69.37 | 52.72 | 23.50 | 38.81 | 1.72 | 14.63 | 34.30 |

TfQA: Truthful QA, WG: WinoGrande, NQ: Natural Questions, OBQA: OpenBook QA, TrQA:TriviaQA

Table. 5 reports English-language LM evaluation results for our english-only continually pre-trained LLMs. Normalized accuracy is reported for HellaSwag and exact match (EM) is reported for NaturalQuestions and

TriviaQA. All other tasks report unnormalized accuracy. As expected, we observe that the larger (10B) models achieve stronger performance than their smaller counterparts and that models trained on more tokens always achieve better performance than models trained on fewer tokens. For both model scales, we observe that the models pre-trained continually using a combination of learning rate re-warming and 5% replay approach (10B) or surpass (405M) the performance of the models trained on the union of both datasets in terms of average accuracy. When comparing union-trained models to continually pre-trained models for different tasks, we observe for the 10B parameter models that the 5% replay model and union-trained model exchange best performance on different tasks with notable differences being OpenBookQA in favor of the replay model and MMLU in favor of the union model. While this degradation in MMLU performance between both models could be cause for concern, we suspect it is due to the limited amount of training data used in our study. Following the initial release of this work, Glorioso et al. (2024) successfully applied our techniques without MMLU performance degradation; in fact, their performance on MMLU is improved during continual pre-training. For the 405M parameter models, the 5% replay model and union-trained model exchange best performance on different tasks with no notable differences. At both model scales, the replay model improves over the model only using re-warming though differences are small and may be attributable to noise.

*In summary, we find that models continually pre-trained with a combination of LR re-warming, LR re-decaying, and replay exceed the average performance (e.g., w.r.t. final validation loss and evaluation accuracy) of baselines trained from random initialization on individual datasets and achieve comparable evaluation performance on average to the expensive re-training baseline (trained on the union of both datasets). These results show that the benefits of continual pre-training hold at the $10B$ parameter scale, suggesting that this may also be the case for models with an order of magnitude more parameters (e.g. for $100B+$ parameters).*

# 7 Understanding and Circumventing the Pathologies of Re-warming

In this section, find that LR re-warming causes unwanted forgetting, introduce infinite learning rate schedules as a promising way to circumvent it, and compare these schedules to baselines from the literature.

## 7.1 Re-warming on the Same Data

In section 6.1, we have seen that continuing to pre-train on new data initially leads to a quick increase of the loss on past data, which motivated the use of replay. The increase of the loss was, in particular, more pronounced for greater $\eta_{max}$ values. One hypothesis for the increase in loss is that it is mostly due to a distribution shift between the pre-training datasets and associated negative transfer. To assess this hypothesis, we re-warm and re-decay over 300B tokens in a setting with no distribution shift. That is, we follow a similar methodology as in our experiments from Fig. 4 but continue to pre-train on Pile as $\mathcal{D}_1$.

As seen in Fig. 8, independently of the distribution shift, rewarming the learning rate appears to be a significant cause of the increase in loss seen previously in Fig. 4 when starting to continue to pre-train, as evidenced by the increase in perplexity when re-warming the learning rate while training on the same distribution. For example, the re-warming leads to a peak increase of the Pile validation loss of 0.1 relative to its initial value with a $\eta_{max} = 3 \cdot 10^{-4}$ as we continue pre-training on Pile, which might be contrasted with the Pile validation loss increase of 0.35 with the same learning rate schedule when continuing to pre-train on SlimPajama as in Fig. 4. It is noteworthy that the higher the re-warming, the more pronounced this effect is, as seen with the $\eta_{max} = 6 \cdot 10^{-4}$ curve when continuing to pre-train on Pile (with a peak loss increase of 0.2) vs continuing to pre-train on SlimPajama (peak loss increase of 0.45).

In particular, after re-warming, models fail to recover quickly from the performance hit due to rewarming the learning rate even when training on the same dataset. This motivates finding alternatives to learning rate schedules requiring re-warming in order to improve the efficiency of continual pre-training.

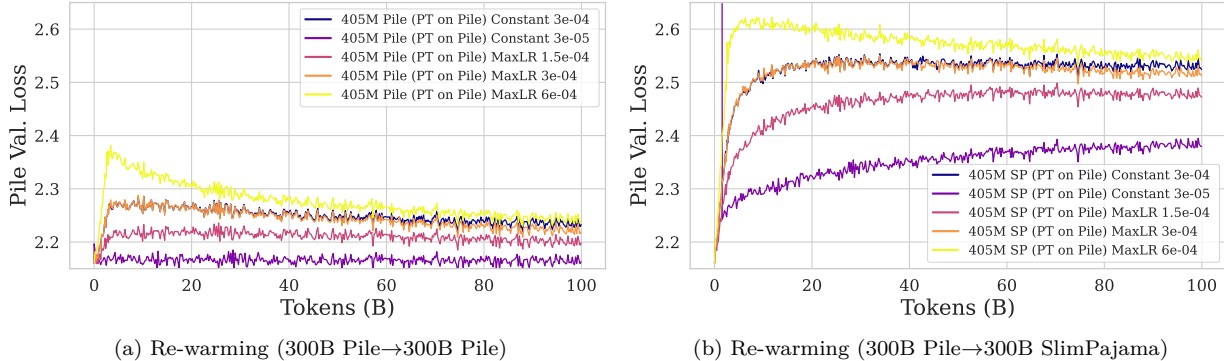

(a) Re-warming (300B Pile→300B Pile)        (b) Re-warming (300B Pile→300B SlimPajama)

Figure 8: **Pile validation loss when continuing to pre-train on Pile (a) and SlimPajama (b).** Each curve starts from the same checkpoint pre-trained on 300B tokens of Pile but is trained with a different maximum learning rate. As we focus on the effect of re-warming the learning rate, we only show curves for the first 100B tokens. We observe that every model that re-increases its learning rate from the minimum learning rate of the initial pre-training (e.g., all models except constant) sees an increase in loss.

## 7.2 Infinite Learning Rate Schedules

In this subsection, we investigate the use of learning rate schedules that intrinsically may not require re-warming. The motivations are twofold. On the one hand, a cosine decay schedule requires us to know the total number of tokens we want to pre-train on in advance. This limits the ability to continue to pre-train a converged checkpoint. On the other hand, we saw in the previous section that when continuing to pre-train a model that was initially pre-trained with a cosine decay schedule ending with a small learning rate, re-warming the learning rate from its minimum value is needed to best adapt to the new dataset. However, as seen in the previous subsection, we observe that re-warming the learning rate can exacerbate forgetting.

Thus, we explore "Infinite Learning rate schedules" (Zhai et al., 2022) which keep the learning rate at a constant value across all new tasks. This can help prevent forgetting by avoiding re-warming the learning on new tasks. Additionally, this schedule is independent of the total number of tokens making it more suitable for continual learning setups compared to repeating the cosine decay schedule cyclically for each new dataset. As we saw, since a high constant learning rate is also suboptimal, we opt to perform a fast annealing of the learning rate at the end of pre-training, over a limited amount of tokens. We hope that this will recover the performance advantage of re-decaying the learning rate, while allowing the use of a pre-annealing checkpoint when continuing to pre-train.

The infinite learning rate schedules considered have 4 phases:

1. **Linear warm-up phase** – As before, the learning rate is initially increased to some maximum value $\eta_{max}$ over $T_{warmup}$ timesteps, or equivalently until timestep $t_{cd} = T_{warmup}$. The learning rate undergoes a warm-up only once (during the first task) and does not require re-warming for future tasks.

2. **Cooldown phase** – During this stage the learning rate undergoes a cooldown phase where the learning rate is gradually decayed to constant value $\eta_{const}$ according to some decay function $f_{cd}$ over $T_{cd}$ timesteps from timestep $t_{cd}$ to $t_{const} = t_{cd} + T_{cd}$. This stage also occurs only once during the first task.

3. **Constant phase** – The learning rate then remains constant for all future tasks over $T_{const}$ timesteps from timestep $t_{const}$ to $t_{ann} = t_{const} + T_{const}$. The checkpoint obtained at the end of this phase is the one one should resume from when continuing to pretrain on a new dataset.

4. **Annealing phase** – The learning rate is annealed to a small value $\eta_{min}$ over $T_{ann}$ timesteps from timestep $t_{ann}$ to $t_{end} = t_{ann} + T_{ann}$, helping train the model to convergence before being deployed.

Thus, the infinite learning rate schedules considered here can be written as:

$$
\eta_t = \begin{cases}
\eta_{max} \cdot \dfrac{t}{T_{warmup}} & t \in [0, t_{cd}] & (warm\text{-}up) \\[2ex]
f_{cd}(t) & t \in (t_{cd}, t_{const}] & (cooldown) \\[1ex]
\eta_{const} & t \in (t_{const}, t_{ann}] & (constant) \\[2ex]
\eta_{const} \cdot \left( \dfrac{\eta_{min}}{\eta_{const}} \right)^{\frac{t - t_{ann}}{t_{end} - t_{ann}}} & t \in (t_{ann}, t_{end}] & (annealing)
\end{cases}
$$

In this work, we consider the two following functions for the cooldown phase's decay $f_{cd}$:

1. Cosine decay

$$
f_{cd}(t) = \eta_{const} + \frac{\eta_{max} - \eta_{const}}{2} \cdot \left( 1 + \cos \left( \pi \left( \frac{t - t_{cd}}{t_{const} - t_{cd}} \right) \right) \right) \tag{3}
$$

2. Inverse Square Root decay

$$
f_{cd}(t) = \eta_{max} + \frac{\eta_{const} - \eta_{max}}{h(1)} \cdot h \left( \frac{t - t_{cd}}{t_{const} - t_{cd}} \right) \tag{4}
$$

where

$$
h(x) = \frac{1}{\sqrt{1 + \alpha x}} - 1
$$

with $\alpha$ controlling the steepness of the inverse square root decay. We shift and stretch the Inverse Square root decay to adapt to the interval $(t_{cd}, t_{const}]$.

The three different schedules are seen in Fig. 9 (b).

We now compare infinite learning rate schedules to a cosine decay schedule. We first explore a simple single-dataset pre-training setup to evaluate the feasibility of the schedule for LLM pre-training. Subsequently, we explore its benefits in our *three datasets, no shift* setting.

### 7.3 Comparing Cosine Decay to Variants of our Infinite Schedules

Here we compare a cosine decay schedule with infinite learning rate schedules in the common single-dataset pre-training setting. The aim of these experiments is to test if the infinite learning rate schedules can result in models that perform as well as models trained with a conventional cosine decay schedule.

The models are pre-trained on 300B tokens of SlimPajama from random initialization. Figure 9 shows the training curves of 3 405M parameter models trained on SlimPajama with different learning rate schedules. We observe that all methods reach similar final validation loss showing that infinite learning rate schedules can be used for the common case of pre-training as well. These schedules additionally have the advantage that one can start annealing at any time in the constant phase to efficiently improves the loss when deciding to finalize pre-training, and a pre-annealing checkpoint can be loaded to continue pre-training.

### 7.4 Infinite Learning Rate Schedules: Scaling to Infinite Future Updates

We now explore the role of the infinite learning rate schedules when multiple new datasets are seen in a continual learning setup. The models are trained from random initialization with different learning rate schedules on 3 IID 100B subsets of SlimPajama (e.g., our *three datasets no shift* setting; see Sec 5.2). We focus on the no shift setting in these preliminary experiments and leave the weak and strong shift cases to future work. This task simulates a setting where large amounts of data from the same distribution are received at time increments and we wish to continue pre-training our models on them (e.g., continuing to pre-train the model on the latest web-scrape). To make our results applicable to situations where previous

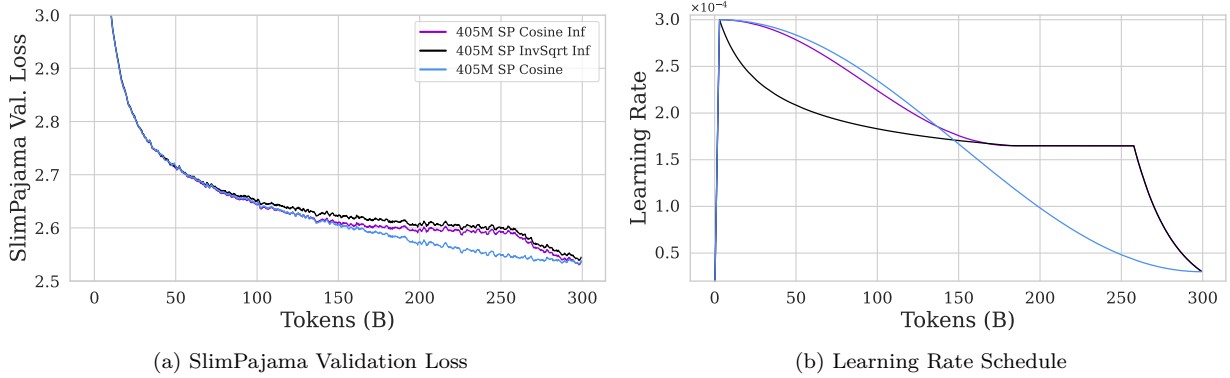

(a) SlimPajama Validation Loss

(b) Learning Rate Schedule

Figure 9: **Infinite learning rate schedules v.s. Cosine decay.** We train a 405M parameter model on 300B tokens of SlimPajama from random initialization with two new schedules, *Cosine Inf* and *InvSqrt Inf*, and compare them to the cosine decay baseline. *Cosine Inf* and *InvSqrt Inf* first decay to a fixed constant LR value and stay constant thereafter until an abrupt final decay. These schedules, therefore, have the advantage that they can smoothly transition between one pre-training phase and the next without re-warming. We find that all methods reach similar final validation loss showing that Cosine decay is not a prerequisite for strong performance.

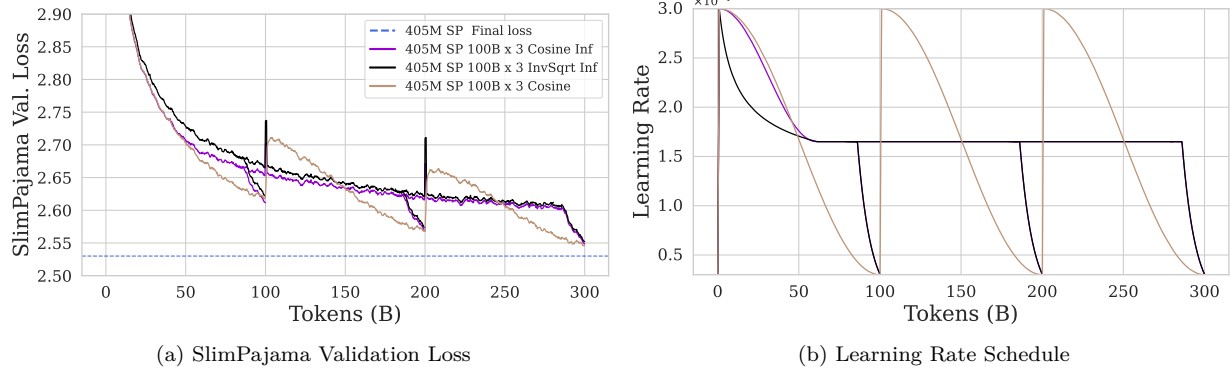

(a) SlimPajama Validation Loss

(b) Learning Rate Schedule

Figure 10: **Infinite learning rate schedules evaluated on 3 IID 100B token subsets of SP.** The experiment simulates a setting where new data from the same distribution arrives over time and the practitioner wishes to update their model on the new data. The models are trained from random initialization on the first dataset. For each dataset, we train two checkpoints: a checkpoint that continues the constant phase for all data in this dataset and a decayed checkpoint (e.g., phase 4). When transitioning to the new datasets, we select the former. We note that, in figure (b), the black and violet schedules overlap after $\sim 80$B tokens.

optimizer states are not available, we do not keep optimizer states across dataset boundaries. Fig. 10 reports training curves for 405M parameter models.

We observe that all schedules perform relatively similarly, however, the two infinite schedules have the advantage that we can start annealing at any time during the constant learning rate phase on each split, while the repeated cosine decays require knowing the number of tokens in advance. Additionally, we see negligible forgetting across dataset boundaries for the infinite LR schedules. While the losses initially increase sharply due to re-initializing the optimizer states, the infinite schedules models immediately recover from this.

In future works, it would be interesting to study the impact of infinite learning rate schedules in continual learning setups with distribution shifts, and investigate the stability of training over large amounts of tokens with a long constant phase of the learning rate.

*In summary, we saw that re-warming can hurt performance even when training on the same distribution, but that alternatives to cosine decay schedules might circumvent these issues. Furthermore, these infinite learning rate schedules provide a simple way to end or resume pre-training without being constrained to a particular token budget. That being said, settings with distribution shifts should also be explored to validate these schedules.*

## 8    Limitations

While we have conducted a thorough empirical evaluation of continual pre-training for LLMs, there are some limitations to our work. In no particular order: 1) we only studied two model sizes (405M and 10B); 2) we did not run deduplication between the German training and validation datasets created from the German Common Crawl scrape (Laippala et al., 2022); 3) we primarily study the transition between two subsequent tasks; 4) we did not run our experiments over multiple seeds; and 5) our experiments on infinite learning rate schedules are limited to 405M scale with no distribution shift. More explicitly, the first limitation is the number of model scales we consider. While we do consider a 405M and a 10B parameter model (much larger than most works), we could not extend the study to another order of magnitude due to computational limitations (e.g., 100B parameter scale). The second limitation of our work is that the German validation set was not deduplicated from the German training data. While we were careful to take distinct shards for training and validation, there may be some contamination between the two. Given that all baselines have access to the same dataset, however, we believe our results are still valid. The third limitation is that we did not run experiments updating models on more than two subsequent tasks. While we believe that studying this is important, our goal was to focus our compute on different distribution shifts and studying transitions between large datasets, rather than using a large number of datasets. The fourth limitation is that we did not run experiments over multiple seeds due to high computational cost, meaning that there is likely a stochastic element to some results. That being said, our LLMs are trained with a large batch size (2M+ tokens) and, thus, there is little variance in the gradient estimates. Coupled with the fact that the samples from each dataset are processed in the same order in all cases, we believe that our results should be relatively stable to changes in random initialization dictated by the seed. The fifth limitation is that it is very possible that over enough tokens, the infinite schedules may end up being suboptimal due to only having a single phase of warmup and cooldown, as the learning on all subsequent datasets may just be equivalent to using a constant learning rate, which proved to be suboptimal (see Fig. 4). While Fig. 10 showed that the annealing phase helps recover from this suboptimality in the case of IID splits of the same dataset, it is unclear if this would hold over more tokens, or in the case where the different datasets have distribution shifts. Hence, experiments involving distribution shifts, and a larger scale of models and datasets would be important to further test these infinite schedules. Finally, another important consideration to explore at a larger scale is the stability of pre-training with such schedules (in particular, during the constant learning rate phase without $\mu P$ (Yang et al., 2022)).

## 9    Conclusion

In the context of continual pre-training of autoregressive transformer-based LLMs, we have seen that learning rate re-warming and re-decaying is important for adaptation and found that forgetting is easily mitigated with replay in this setting—at seemingly little cost to adaptation. Given their powerful ability to enhance adaptation and mitigate forgetting simultaneously, we proposed the simple and scalable combination of LR re-warming, LR re-decaying, and replay for continually pre-training LLMs at scale. We showed that these strategies enable continual pre-training to achieve average performance on par with expensively re-training from scratch on all data, across two distribution shifts (weak & strong) and two decoder-only transformer LLM scales (405M & 10B). Upon further analysis, we identified a pathology of LR re-warming and, inspired by previous work, proposed infinite learning rate schedules for continually pre-training LLMs. In initial experiments, our schedules achieve performance on par with cosine decay while circumventing the need for LR re-warming.

Our findings show that continual pre-training is an efficient and promising alternative to re-training when updating decoder-only transformer LLMs on new data. Equipped with our strategies, practitioners can

efficiently update their existing models (Rae et al., 2021; Hoffmann et al., 2022; Touvron et al., 2023b; Jiang et al., 2023; Gemma Team et al., 2024) on newly created higher-quality datasets. These strategies might also be relevant for pre-training curricula such as the ones used by Gemma Team et al. (2024). With the strong incentive for our community to continue creating datasets of increasing quality, we only expect the need for continual pre-training to increase.

In follow-up work, it will be important to further investigate infinite learning rate schedules, growing models during continual pre-training (e.g., mixture-of-experts or block expansion), and adapting the tokenizer to handle drastic changes to the data distribution. Moreover, we would like to explore continual pre-training in the context of multimodal or vision language models and other text-based generative models—we note that recently, Garg et al. (2023) concurrently replicated the success of the techniques discussed in this work in the context of CLIP models instead of LLMs. We also would like to explore replay buffer creating in the continual pre-training setting where an open-weight model does not disclose its dataset; we suspect using the available model for synthetic data or distillation may be a promising direction to build the replay buffer.

## Broader Impact Statement

Large language models have seen widespread adoption across a wide range of industry sectors due to their ability to perform very well after being trained on relevant datasets. Moreover, improvements in datasets (better filtering, updating knowledge, etc.) have been crucial to increasing the quality of the output of LLMs. As such, it is reasonable to expect that organizations will spend a significant amount of computing power and, thus, energy to create more powerful models. It is likely that some of this energy will come from non-renewable sources. While the experiments presented in our paper are environmentally costly, as argued in the paper, continuing to pre-train is a promising method to significantly reduce the compute associated with updating a model and, hence, the energy required to maintain foundation models.

## Acknowledgements

We acknowledge support from NSERC Discovery Grant RGPIN- 2021-04104 [E.B.], the Canada CIFAR AI Chair Program [I.R.], and the Canada Excellence Research Chairs Program [I.R.]. We would also like to acknowledge funding from the FRQNT Doctoral (B2X) scholarship [B.T.], the scholarship for Artificial Intelligence of Université de Montréal's Études Supérieures et Postdoctorales [A.I.], and a fellowship of the IFI program of the German Academic Exchange Service (DAAD)[M.R.]. This research was made possible thanks to the computing resources on the Summit supercomputer, provided as a part of the INCITE 2023 program award "Scalable Foundation Models for Transferable Generalist AI". These resources were provided by the Oak Ridge Leadership Computing Facility at the Oak Ridge National Laboratory, which is supported by the Office of Science of the U.S. Department of Energy under Contract No. DE-AC05-00OR22725. In particular, we thank Jens Glaser for his help with the Summit supercomputer.

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

## Contents

# Appendix

## A    Extended results

In the following subsections, we first present some new results in the *Domain incremental continual pre-training* setting, comparing replay for different dataset sizes, and providing a qualitative analysis of our German language models. We also provide aggregated evaluation and final loss tables for all models in the paper.

### A.1    Domain Incremental continual pre-training

We consider a **domain incremental learning** setting, where we train on a sequence of $N$ future datasets $\{\mathcal{D}_0, \mathcal{D}_1, \ldots, \mathcal{D}_{N-1}\}$ each of which comes from a distinct domain. This setting is particularly challenging due to the distribution shift experience at the transition between each domain. Concretely, we consider dataset $\mathcal{D}_0$ to be pre-training on the Pile (Gao et al., 2020) and $\mathcal{D}_1$ to be pre-training on SlimPajama. However, instead of consuming the data in $\mathcal{D}_1$ using the sampling percentages from table 1, we process the dataset one domain at a time starting from the largest domain to the smallest. This simulates a situation where new data is received from different domains at different times and we wish to update our models on the sequence while being robust to all distribution shifts.

**Adapting replay to Domain Incremental continual pre-training**    In settings that span more than two tasks, we use a form of reservoir sampling (Buzzega et al., 2020), where the replay buffer is updated at discrete intervals. We refer to this technique as *discrete reservoir sampling.* Specifically, given a sequence of $N$ datasets $\mathcal{D}_0, \mathcal{D}_1, \ldots, \mathcal{D}_{N-1}$ of sizes $s_0, s_1, \ldots, s_{N-1}$ that are trained on in sequence, we update the replay buffer $\mathcal{R}$ at each dataset transition. Let $\mathcal{R}_i$ correspond to the state of the replay buffer as the start of training on the $i$-th dataset. For replay ratio $0 \leq \alpha \leq 1$, at any given $i > 0$, $\mathcal{R}_i$ will contain data from all $\mathcal{D}_j$ for $j < i$ in proportions

$$p_{i,j} := \frac{s_j \cdot (1-\alpha)^{\gamma_j} + \sum_{k=j+1}^{i-1} p_{k,j} \cdot s_k \cdot \alpha}{\sum_{k=0}^{i-1} s_k} \quad \text{with} \quad \forall j, \ p_{j+1,j} = 1, \tag{5}$$

where $i$ is the index of the dataset on which we are currently pretraining, and $\gamma_j = 0$ if $j = 0$, and 1 otherwise. The latter is because when pretraining on $\mathcal{D}_i$, we only see $s_i \cdot (1-\alpha)$ tokens of $\mathcal{D}_i$ because we use compute-equivalent replay, except for the first dataset $\mathcal{D}_0$ where replay is not used, and where we hence see all $s_0$ tokens of $\mathcal{D}_0$.

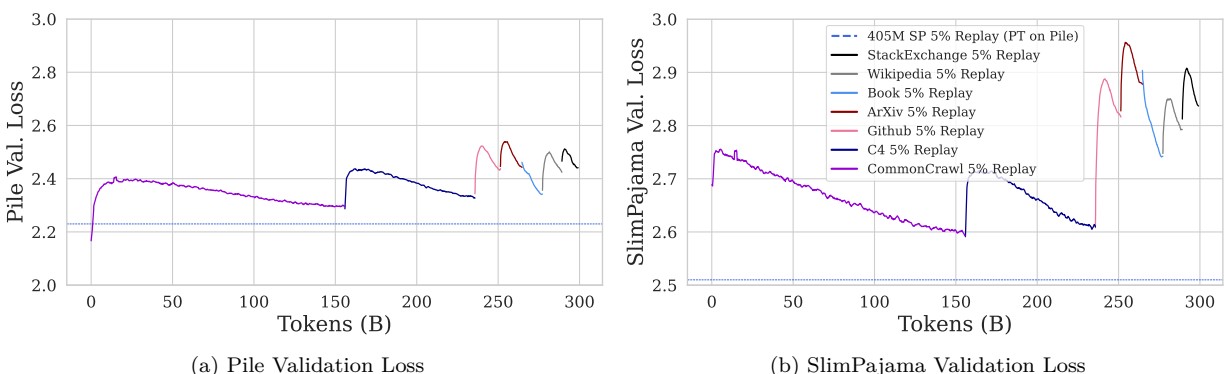

(a) Pile Validation Loss          (b) SlimPajama Validation Loss

Figure 11: **Ingesting data sequentially by domain.** We explore an alternative approach to consuming the available data. Similar to task-incremental or class-incremental learning, we train on one domain at a time in a sequential fashion. We use LR re-warming and LR re-decaying for each domain as well as *discrete reservoir sampling.*

Figure 11 reports results for a domain incremental approach to consuming our 300B token split of SlimPajama, where each domain is processed in decreasing order of size. We use learning rate re-warming, LR re-decaying, and replay with discrete reservoir sampling. The total data processed is the same as the model trained with LR re-warming and re-decaying on the 300B tokens of SlimPajama (blue dotted line). However, the order in which the data is processed and the learning rate schedule differs from the baseline.

We hypothesize that difficulties arise due to relatively strong distribution shifts and the many learning rate re-warmings. The number of consecutive distribution shifts with varying amounts of data from different domains makes it difficult to obtain strong final performance across all of them. In particular, it seems especially difficult to efficiently adapt to the new data distribution. We hypothesize part of these difficulties are optimization-related. Less frequent warming up and decaying of the learning rate may be beneficial in this setting, and the infinite learning rate schedules might help, but future research is needed to answer this question. At any rate, we believe this setting is unnecessarily difficult compared to consuming the data using a mixture (as we did in the main paper) and is therefore of limited practical use.

## A.2   Model Merging v.s. Continual Pre-training

Some may be curious about the performance of model merging as a continual pre-training approach, and how it compares to continuing to pre-train with the methods used in this paper. In preliminary experiments with TIES (Yadav et al., 2023b) using Mergekit v0.0.4 (Goddard et al., 2024), we consider merging models trained separately on each dataset (e.g., merging 300B SP into 300B Pile and 200B German into 300B Pile) using 300B Pile as the base model. We also merge continually pre-trained checkpoints into 300B Pile. English and German LM evaluation results can be found in Tables 6 and 7.

We find that, in general, continual pre-training outperforms merging models trained separately on each dataset and requires the same amount of compute. Moreover, we find that merging checkpoints of a continually pre-trained model does not increase performance, likely due to replay being a more efficient strategy. However, we note that these experiments are by no means comprehensive; other techniques may yield better performance.

Table 6: **English LM Eval Performance of Models Merged with TIES v.s. Continual Pre-training.** Normalized accuracy is reported for HellaSwag and exact match (EM) is reported for NaturalQuestions and TriviaQA. All other tasks report unnormalized accuracy. MMLU and TriviaQA are evaluated 5-shot, while all other tasks are zero-shot. In (x% R), the R should be read as Replay.

| Model | Method | Base Model | Task Model(s) | TopK% | Hella-Swag | ARC-c | ARC-e | BoolQ | MathQA | MMLU | OpenBookQA | PIQA | WinoGrande | TruthfulQA MC1 | TruthfulQA MC2 | NaturalQuestions (EM) | TriviaQA (EM) | AVG |
|---|---|---|---|---|---|---|---|---|---|---|---|---|---|---|---|---|---|---|
| 405M | TIES | 300B Pile | 200B Ger. | 25 | 25.91 | 20.31 | 24.87 | 40.83 | 18.46 | 26.06 | 15.20 | 54.03 | 50.28 | 24.60 | 50.48 | 0.00 | 0.01 | 27.00 |
| | | | | 75 | 27.64 | 18.00 | 30.43 | 42.57 | 21.21 | 25.45 | 15.40 | 55.11 | 51.85 | 24.72 | 45.73 | 0.00 | 0.30 | 27.57 |
| | | | | 50 | 25.97 | 19.62 | 27.36 | 37.98 | 19.20 | 26.41 | 15.60 | 53.75 | 51.14 | 21.91 | 47.01 | 0.00 | 0.00 | 26.61 |
| | | | 300B Pile → 200B Ger. (25% R) | 25 | 30.62 | 19.11 | 35.61 | 62.11 | 21.57 | 25.37 | 14.80 | 59.19 | 50.75 | 24.60 | 44.81 | 0.08 | 0.51 | 29.93 |
| | | | | 50 | 34.52 | 20.90 | 42.63 | 61.77 | 23.48 | 25.02 | 17.00 | 61.37 | 50.59 | 26.56 | 45.78 | 0.25 | 2.65 | 31.73 |
| | | | | 75 | 35.78 | 20.99 | 45.62 | 61.99 | 23.02 | 25.25 | 18.60 | 62.89 | 51.78 | 25.83 | 43.84 | 0.42 | 5.78 | 32.44 |
| | Baseline | | 300B Pile → 200B Ger. (25% R) | – | 36.05 | 21.59 | 47.56 | 60.49 | 23.82 | 25.00 | 17.20 | 63.49 | 51.14 | 25.70 | 43.84 | 0.36 | 5.97 | 32.48 |
| | TIES | 300B Pile | 300B SP | 25 | 25.14 | 21.67 | 25.88 | 37.83 | 19.87 | 24.65 | 16.00 | 52.12 | 50.67 | 24.36 | 48.78 | 0.03 | 0.02 | 26.69 |
| | | | | 50 | 26.31 | 19.97 | 28.70 | 37.83 | 20.97 | 24.56 | 12.40 | 54.52 | 53.51 | 23.50 | 51.25 | 0.06 | 0.02 | 27.20 |
| | | | | 75 | 41.35 | 19.80 | 48.11 | 60.06 | 22.21 | 25.10 | 16.00 | 65.94 | 48.93 | 22.03 | 40.57 | 0.80 | 6.36 | 32.10 |
| | | | 300B Pile → 300B SP (5% R) | 25 | 32.40 | 19.97 | 35.02 | 51.74 | 21.41 | 24.10 | 14.20 | 58.54 | 52.17 | 24.97 | 46.02 | 0.17 | 0.44 | 29.32 |
| | | | | 50 | 38.98 | 21.33 | 45.71 | 44.62 | 23.12 | 24.11 | 16.60 | 63.55 | 50.99 | 25.46 | 44.00 | 1.16 | 3.98 | 31.05 |
| | | | | 75 | 44.80 | 23.21 | 53.79 | 58.84 | 23.69 | 24.84 | 19.40 | 68.12 | 52.96 | 23.01 | 39.24 | 2.52 | 13.02 | 34.42 |
| | Baseline | | 300B Pile → 300B SP (5% R) | – | 46.55 | 23.55 | 55.01 | 57.92 | 24.22 | 25.94 | 20.60 | 69.37 | 54.22 | 23.38 | 38.35 | 1.99 | 15.70 | 35.14 |
| 10B | TIES | 300B Pile | 300B SP | 25 | 25.45 | 22.18 | 25.63 | 52.45 | 18.66 | 26.93 | 16.20 | 53.86 | 48.15 | 23.75 | 48.82 | 0.00 | 0.01 | 27.85 |
| | | | | 50 | 31.85 | 21.16 | 35.56 | 62.17 | 19.46 | 24.08 | 16.60 | 60.50 | 51.14 | 24.24 | 45.39 | 0.08 | 0.01 | 30.17 |
| | | | | 75 | 66.93 | 33.02 | 66.75 | 66.02 | 24.59 | 27.03 | 25.00 | 74.21 | 62.83 | 21.91 | 34.71 | 6.90 | 35.34 | 41.94 |
| | Baseline | | 300B Pile → 300B SP (5% R) | – | 73.24 | 39.42 | 74.24 | 70.80 | 26.83 | 28.79 | 30.60 | 78.02 | 68.67 | 23.01 | 35.02 | 13.32 | 57.86 | 47.68 |

TfQA: Truthful QA, WG:WinoGrande, NQ: Natural Questions, OBQA: OpenBook QA, TrQA:TriviaQA

## A.3   Replay for Different Dataset Sizes

To ablate the insights gathered from the experiments in Sec. 6.2 we vary the size of the continued pre-training on SlimPajama with 100B, 200B, and 300B tokens. The respective linear warmup schedules and cosine decays are fitted to the altered length of training. As expected, we observe a consistent decrease in loss at the end of training for SlimPajama with an increased number of training tokens (see Fig. 12). We further observe similar trends concerning the effect of replay on forgetting and adapting to novel data. In conclusion, the simple combination of LR re-warming, LR re-decaying, and replay is effective at different dataset sizes.

Table 7: **German LM Eval Performance of Models Merged with TIES v.s. Continual Pre-training.** Normalized accuracy is reported for HellaSwag and exact match (EM) is reported for NaturalQuestions and TriviaQA. All other tasks report unnormalized accuracy. MMLU and TriviaQA are evaluated 5-shot, while all other tasks are zero-shot.

| Model Size | Merging Method | Base Model | Task Model(s) | TopK% | MMLU DE | ARC-C DE | Hella-Swag DE | TruthfulQA DE | AVG |
|---|---|---|---|---|---|---|---|---|---|
| 405M | TIES | 300B Pile | 200B Ger. | 25 | 22.86 | 20.65 | 24.78 | 21.05 | 22.33 |
| | | | | 75 | 24.38 | 19.54 | 28.67 | 25.83 | 24.60 |
| | | | | 50 | 23.47 | 21.08 | 25.64 | 23.50 | 23.42 |
| | | | 300B Pile → 200B Ger. (25% Replay) | 25 | 23.89 | 19.45 | 26.76 | 25.70 | 23.95 |
| | | | | 50 | 25.12 | 18.86 | 29.45 | 26.32 | 24.94 |
| | | | | 75 | 23.91 | 18.94 | 30.81 | 24.60 | 24.57 |
| | Baseline | 300B Pile → 200B Ger. (25% Replay) | | – | 23.78 | 19.20 | 31.04 | 25.58 | 24.90 |

TfQA: Truthful QA, WG:WinoGrande, NQ: Natural Questions, OBQA: OpenBook QA, TrQA:TriviaQA

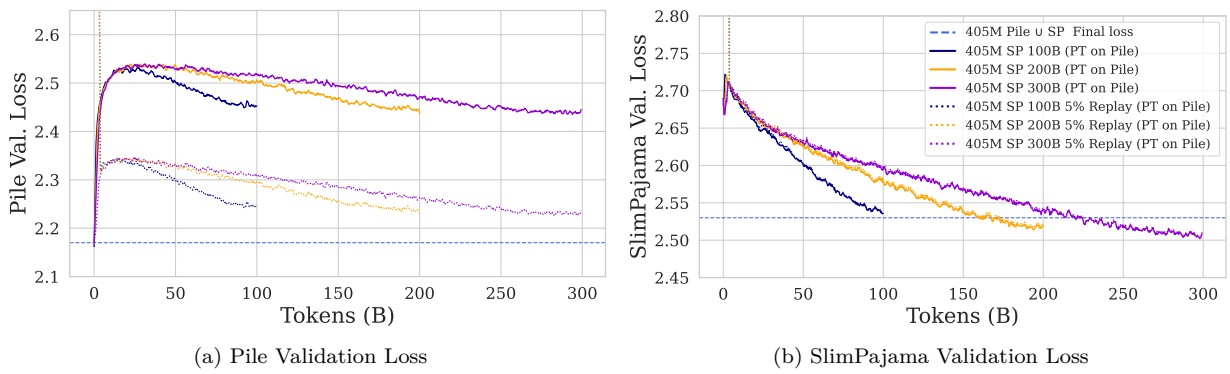

(a) Pile Validation Loss            (b) SlimPajama Validation Loss

Figure 12: **Replay v.s. no replay when re-warming the learning rate and continually pre-training on different amounts of data.** We train 405M parameter models with and without 5% replay on 100B, 200B, and 300B tokens of SlimPajama data. Each model starts from a checkpoint pre-trained on 300B tokens of Pile and re-warms the learning rate using a linear warmup and cosine decay schedule fitted to its dataset size.

## A.4 The effect of resetting optimizer states at dataset transitions

In all of our experiments, we drop optimizer states between dataset transitions to show that our techniques can be used to continually pre-training open-source LLMs, which rarely provide optimizer states. In this section, we investigate whether keeping the optimizer states from pre-training can resolve instabilities seen during continual pre-training. Figure 13 reports validation loss curves during the continual pre-training of a 405M parameter decoder-only transformer on 40B tokens of SlimPajama data. Each model starts from a checkpoint pre-trained on 300B tokens of Pile and re-warms and re-decays the learning rate using a linear warmup and cosine decay schedule fit to a dataset size of 300B tokens ($\eta_{max} = 3 \cdot 10^{-4}$, $\eta_{min} = 3 \cdot 10^{-5}$). We observe that, initially, both models follow slightly different trajectories. However, after 40B tokens of training both models reach similar loss values with differences attributable to randomness. We therefore conclude that, in this setting, keeping optimizer states does not affect continual pre-training compared to discarding them. We note that this is to be expected, as the relatively large momentum coefficient values used for LLM pre-training (0.9 and 0.95 for $\beta_1$ and $\beta_2$) cause the contribution of the starting optimizer states to quickly decay to 0. For instance, after 1000 steps of optimization, the moment estimates from pre-training will contribute to $< 0.0044\%$ of the current moment estimates.

## A.5 Qualitative evaluation of German models

In this section, we provide a brief qualitative evaluation of the models trained on German Common crawl (Sec. 6.3). We select five German prompts that contain various peculiarities of the German language (see Tab. 8). We then generate a fixed token-length response for each of the models trained or continually

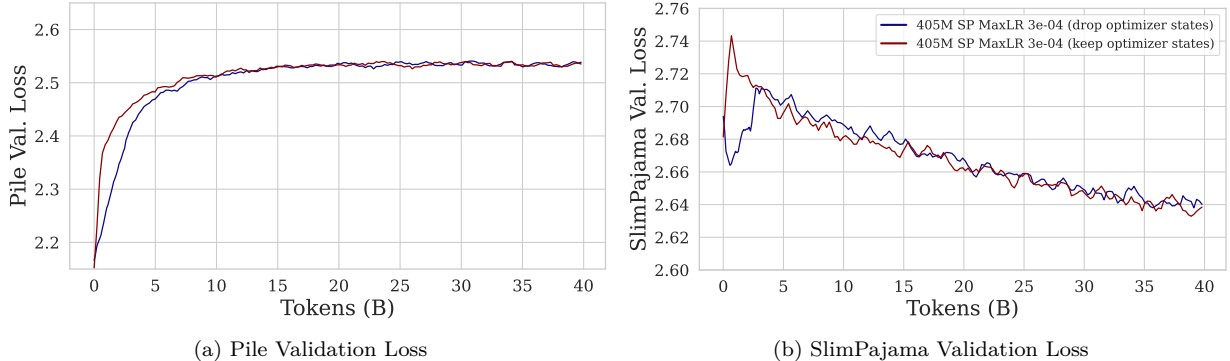

(a) Pile Validation Loss

(b) SlimPajama Validation Loss

Figure 13: **Keeping v.s. dropping optimizer states when transitioning from Pile to Slim Pajama.** We train a 405M parameter decoder-only transformer on 40B tokens of SlimPajama data. Each model starts from a checkpoint pre-trained on 300B tokens of Pile and re-warms and re-decays the learning rate using a linear warmup and cosine decay schedule fitted to a dataset size of 300B tokens ($\eta_{max} = 3 \cdot 10^{-4}$, $\eta_{min} = 3 \cdot 10^{-5}$). We observe that, initially, both models follow slightly different trajectories. However, after 40B tokens of training both models reach similar loss values with differences attributable to randomness.

pre-trained on German Common Crawl. As a baseline, we also evaluate the same model trained only on the Pile.

We provide the responses and a manual translation of the response in Table 9. While the relatively small 405M parameter models generally do not produce meaningful sentences, the model trained on German Common Crawl generally produces grammatically correct sentences. This includes correct upper and lower casing, the generation of compound words, and the use of Umlaut characters. In all cases, responses tend to be repetitive after a few words. The incompleteness and fractional words of some outputs can be attributed to the number of tokens generated being very short. Longer sequences tend to provide syntactically correct sentences but also tend to repeat themselves. The limited number of samples and the generally low quality of the generated text does not allow us to make robust statements of the qualitative differences between the individual models trained on German Common Crawl. However, the aforementioned problems of repetitiveness as well as grammatical errors seem to be significantly stronger on the Pile baseline, which generally fails to even respect the context given in the prompt. This is expected as the amount of German text is significantly smaller on the Pile.

Thus we conclude that there is an observable effect on the generative quality of the German language from the models that used the data of German Common Crawl during training.

Table 8: German phrases used for qualitative evaluation alongside their translation (for readability) and justification.

| Ref. | Prompt | Translation | Reasoning |
|---|---|---|---|
| $P_1$ | Eine typisch deutsche Mahlzeit ist | A typical German meal is | Conversational sentence. |
| $P_2$ | Döner ist besser als Poutine weil | Doener is superior to poutine because | This prompt contains an Umlaut. |
| $P_3$ | Das Pferd isst keinen Gurkensalat. | The horse does not eat cucumber salad | Short sentence with compound word |
| $P_4$ | Handy ist ein deutsches Wort | Cellphone is a german word | Prompt with an anglicism (Handy = Cellphone) |
| $P_5$ | Der Eiersollbruchstellenverursacher ist | The egg breaker is | Long compound word |

Table 9: Generated text from various prompt-model combinations. All text was generated with a fixed token length and thus is sometimes incomplete. A manual translation of the text is also provided to give the non-German speaker an idea of the generative quality. The translation is as literal as possible to allow for adequate assessment.

| Prompt Ref. (translation) | Model | Fixed Length Response | Translation |
|---|---|---|---|
| $P_1$: (A typical German meal is) | 405M 500B Pile ∪ SP | ein Apfelkuchen.
Wie viel kostet es, einen Apfelk | an apple pie. How much does
an [Apple pie incomplete] cost |
| | 405M 200B German | ein Gericht, das in der Regel .
aus mehreren Gängen besteht | a dish, that usually
consists of multiple courses. |
| | 405M 300B Pile → 200B German (25% Replay) | das Bier.
Wie wäre es mit einem leckeren Eis? | Beer.
Would you like some delicious ice cream? |
| | 405M 300B Pile → 200B German | der Kartoffelsalat.
Kartoffelsalat | the potato salad
potato salad |
| | 405M 300B Pile | die Kinderwunsch
Die Kinderwunsch ist eine der wicht | the desire to have kids
The desire to have kids is the most [important incomplete] |
| $P_2$: Doener is superior to poutine because | 405M 500B Pile ∪ SP | viel besser schmeckt.
Wenn du dich für die Küche | tastes much better.
If you are (missing verb) in the kitchen |
| | 405M 200B German | es so lecker ist.
Antworten
Schreibe einen Kommentar An | it is so tasty.
Answer
Write a comment to |
| | 405M 300B Pile → 200B German (25% Replay) | er viel schneller zubereitet ist,
Wie viel kostet ein D | it is much faster to cook.
How much does a D [incomplete] cost |
| | 405M 300B Pile → 200B German | es mehr Fleisch enthält.
Wenn du dich für einen Döner ents | it contains more meat.
If you opt [incomplete] for a Doener |
| | 405M 300B Pile | er eine komplette Kombination
von Kombinationen von Kombinationen von K | he a complete combination of combinations.
of combinations of combinations of K [incomplete] |
| $P_3$: (The horse does not eat cucumber salad) | 405M 500B Pile ∪ SP | Wenn du dich für die Kategorie
habe, dann ist das Pferd ke | If you are interested in the category
then |
| | 405M 200B German | Wenn ich das richtig verstanden
habe, dann ist das Pferd ke | If I understood this correctly, then the horse is
horse is [incomplete] |
| | 405M 300B Pile → 200B German (25% Replay) | Wie kann ich meinem Pferd helfen?
Wenn du dein | How can I help my horse?
If your |
| | 405M 300B Pile → 200B German | Wenn du das Pferd nicht magst, dann kanns
du es auch mit einem | If you don't like the horse,
then you can also [missing verbt] with a |
| | 405M 300B Pile | Die Pferde sind nicht nur die
Pferde, sondern auch die Pfer | The horses are not only the horses,
but also the horses (last word incomplete) |
| $P_4$: Cellphone is a german word | 405M 500B Pile ∪ SP | das aus dem Lateinischen stammt.
Wie funktioniert das mit dem Handy? | that originates from latin.
How does this work with the cellphone? |
| | 405M 200B German | das in der deutschen Sprache für
das Handy steht. Es ist ein Wort, | that refers to the cellphone in German.
It is a word, [incomplete] |
| | 405M 300B Pile → 200B German (25% Replay) | das für das Telefon steht. Es ist
ein Telefon, das mit einem Handy | that translates to telephone. It is a telephone |
| | 405M 300B Pile → 200B German | das aus dem Englischen übersetzt
"Handy" bedeutet. | which translates from English as "cellphone". |
| | 405M 300B Pile | das in der Tat nicht nur für die deutsche Sprache ist,
sondern auch für die deut | which is indeed not only for the German
language, but also for the (last word incomplete) |
| $P_5$:(The egg breaker is) | 405M 500B Pile ∪ SP | der Eiersollbruchstellenverursacher.
Eiersollbruchstellenverurs | the egg breaker. Egg [breaker, incomplete] |
| | 405M 200B German | ein kleiner, aber feiner Unterschied.
Ein Eiersollbru | a small, but important difference,
An egg [breaker, incomplete] |
| | 405M 300B Pile → 200B German (25% Replay) | der Mensch. Wenn wir uns die Eier in den Eierstock legen,
dann ist das ein ganz normaler Vorgang. | the man. When we place the eggs for us in the ovary,
then this is a natural process. |
| | 405M 300B Pile → 200B German | der Eierkocher. Eierkocher
Eierkocher sind | the egg stove. Egg stove
Egg stoves are |
| | 405M 300B Pile | der Einsatz von Einsatzkraftwerken für die
Einsatzfahrze | the use of emergency power plants for the
emergency vehicles emergency [vehicle, incomplete] |

### A.6 Aggregated LM Evaluation Results

We evaluate our models on the following English and German LM evaluation tasks:

1. **HellaSwag** (Zellers et al., 2019) and **HellaSwag DE**: An English commonsense reasoning benchmark composed of multiple-choice questions that are deliberately designed to confuse language models. HellaSwag DE is a German translation of the HellaSwag benchmark.

2. **AI2 Reasoning Challenge (ARC)** (Clark et al., 2018): An English commonsense reasoning benchmark composed of science examination questions in multiple-choice format. The $7,787$ total questions have been divided into an easy subset with $5,197$ questions and a hard subset with $2,590$ questions. ARC-c DE is a German translation of the challenge subset of questions.

3. **BoolQ** (Clark et al., 2019): An English reading comprehension benchmark composed of $15,942$ yes/no question-answering samples. Each example is split into a question, relevant paragraph, and the solution.

4. **MathQA** (Amini et al., 2019): An English math word problem benchmark composed of multiple-choice questions across various areas of mathematics.

5. **MMLU** (Hendrycks et al., 2021) and **MMLU-DE**: An English benchmark designed to evaluate both zero-shot and few-shot scenarios, in order to evaluate both the general knowledge and on-the-fly problem solving of the model under test. MMLU covers a broad range of subjects. MMLU-DE is a German translation of the MMLU question set, which was translated by the OpenAI GPT 3.5 API.

6. **OpenBookQA (OBQA)** (Mihaylov et al., 2018): An English question-answering benchmark modeled after real-world open-book exams for assessing human understanding of a particular subject. Questions about elementary science are paired with scientific facts and common knowledge, which the model is intended to use in multi-hop reasoning.

7. **PIQA** (Bisk et al., 2019): An English question-answering benchmark designed to test the physical commonsense reasoning abilities of the model. Most questions focus on applying uncommon solutions to everyday situations, which requires understanding of the physical world.

8. **WinoGrande** (Sakaguchi et al., 2019): An English natural language understanding benchmark that involves determining when two or more expressions in a text refer to the same entity. The benchmark includes a diverse set of sentences and a new evaluation metric that rewards models for making human-like predictions.

9. **TruthfulQA** and **TruthfulQA DE** (Lin et al., 2022): An English question-answering benchmark designed to evaluate the truthfulness of generated answers to questions. The questions are designed to contain common human misunderstandings that lead to incorrect answers. TruthfulQA DE is a German translation of the TruthfulQA benchmark.

10. **Natural Questions** (Kwiatkowski et al., 2019): Is an English question-answering benchmark consisting of search queries submitted to the Google search engine.

11. **TriviaQA** (Joshi et al., 2017): Is an English question-answering benchmark comprised of question-answer pairs provided by trivia enthusiasts. Its main focus is to determine a model's general world knowledge.

Table 10: **All Zero-shot and Few-shot results on popular LM benchmarks.** Normalized accuracy is reported for HellaSwag and exact match (EM) is reported for NaturalQuestions and TriviaQA. All other tasks report unnormalized accuracy. MMLU and TriviaQA are evaluated 5-shot, while all other tasks are zero-shot. We observe **on average**, as expected, that 10B param. models outperform their 405M counterparts and that the English-only 405M models outperform their German-trained counterparts.

| Model Size | Training Tokens | HellaSwag | ARC-c | ARC-e | BoolQ | MathQA | MMLU | OBQA | PIQA | WG | TfQA1 | TfQA2 | NQ | TrQA | AVG |
|---|---|---|---|---|---|---|---|---|---|---|---|---|---|---|---|
| 10B | 300B Pile | 68.46 | 34.81 | 69.49 | 68.20 | 27.34 | 27.28 | 27.20 | 76.82 | 62.51 | 20.44 | 33.68 | 6.65 | 41.92 | 43.45 |
| | 300B SP | 70.38 | 36.77 | 71.93 | 68.04 | 24.76 | 27.42 | 28.20 | 76.99 | 65.04 | 22.40 | 33.99 | 11.25 | 52.63 | 45.37 |
| | 300B Pile → 300B SP | 73.66 | 37.37 | 73.02 | 73.18 | 26.43 | 29.94 | 30.20 | 78.51 | 66.30 | 23.26 | 35.04 | 12.99 | 57.94 | 47.53 |
| | 300B Pile → 300B SP (5% Replay) | 73.24 | 39.42 | 74.24 | 70.80 | 26.83 | 28.79 | 30.60 | 78.02 | 68.67 | 23.01 | 35.02 | 13.32 | 57.86 | 47.68 |
| | 600B Pile ∪ SP | 73.39 | 39.25 | 73.57 | 72.05 | 26.83 | 37.78 | 27.80 | 77.58 | 67.32 | 23.13 | 36.16 | 12.41 | 56.73 | 48.00 |
| 405M | 300B Pile | 40.95 | 22.01 | 51.77 | 59.24 | 24.12 | 26.18 | 19.80 | 66.59 | 53.83 | 24.85 | 42.11 | 0.91 | 8.97 | 33.95 |
| | 300B SP | 44.22 | 21.76 | 54.08 | 59.63 | 22.71 | 26.18 | 19.60 | 68.23 | 49.80 | 22.64 | 38.63 | 1.69 | 14.18 | 34.11 |
| | 300B Pile → 300B SP | 46.22 | 22.70 | 54.04 | 57.43 | 24.22 | 25.28 | 21.20 | 69.26 | 54.46 | 23.13 | 38.91 | 2.02 | 15.23 | 34.93 |
| | 300B Pile → 300B SP (0.5% Replay) | 46.26 | 21.42 | 55.18 | 60.18 | 24.09 | 25.90 | 19.80 | 68.72 | 55.64 | 23.38 | 39.13 | 1.86 | 15.58 | 35.16 |
| | 300B Pile → 300B SP (1% Replay) | 46.09 | 23.55 | 54.88 | 57.95 | 23.42 | 26.02 | 20.60 | 68.66 | 54.78 | 22.89 | 37.83 | 1.91 | 15.29 | 34.91 |
| | 300B Pile → 300B SP (5% Replay) | 46.55 | 23.55 | 55.01 | 57.92 | 24.22 | 25.94 | 20.60 | 69.37 | 54.22 | 23.38 | 38.35 | 1.99 | 15.70 | 35.14 |
| | 300B Pile → 300B SP (10% Replay) | 46.15 | 21.93 | 54.42 | 58.44 | 23.62 | 25.53 | 21.40 | 68.77 | 54.54 | 23.75 | 37.87 | 2.38 | 15.00 | 34.91 |
| | 300B Pile → 300B SP (50% Replay) | 44.77 | 23.55 | 53.75 | 60.49 | 25.03 | 26.04 | 20.00 | 68.72 | 53.91 | 23.50 | 39.69 | 1.14 | 13.67 | 34.94 |
| | 600B Pile ∪ SP | 45.06 | 23.55 | 52.99 | 55.57 | 23.12 | 26.65 | 18.20 | 69.37 | 52.72 | 23.50 | 38.81 | 1.72 | 14.63 | 34.30 |
| | 200B Ger. | 27.47 | 17.15 | 29.80 | 45.11 | 21.11 | 25.27 | 13.40 | 56.75 | 52.17 | 26.07 | 46.02 | 0.03 | 0.31 | 27.74 |
| | 300B Pile → 200B Ger. | 30.02 | 19.20 | 32.74 | 61.80 | 20.40 | 23.34 | 13.40 | 58.05 | 49.96 | 24.48 | 44.01 | 0.19 | 1.93 | 29.20 |
| | 300B Pile → 200B Ger. (1% Replay) | 30.84 | 18.09 | 36.49 | 57.80 | 20.60 | 25.28 | 14.60 | 59.52 | 49.49 | 24.72 | 45.74 | 0.11 | 2.81 | 29.65 |
| | 300B Pile → 200B Ger. (5% Replay) | 32.88 | 21.25 | 41.12 | 60.06 | 22.58 | 24.94 | 15.80 | 62.35 | 51.30 | 25.46 | 45.36 | 0.11 | 4.62 | 31.37 |
| | 300B Pile → 200B Ger. (10% Replay) | 34.10 | 20.65 | 43.73 | 52.60 | 22.35 | 25.41 | 18.40 | 63.06 | 50.04 | 25.34 | 44.33 | 0.19 | 4.59 | 31.14 |
| | 300B Pile → 200B Ger. (25% Replay) | 36.05 | 21.59 | 47.56 | 60.49 | 23.82 | 25.00 | 17.20 | 63.49 | 51.14 | 25.70 | 43.84 | 0.36 | 5.97 | 32.48 |
| | 300B Pile → 200B Ger. (50% Replay) | 38.38 | 20.65 | 49.54 | 60.09 | 24.12 | 26.45 | 18.60 | 65.78 | 50.51 | 25.95 | 42.47 | 0.69 | 7.87 | 33.16 |
| | 600B Pile ∪ Ger. | 36.99 | 19.37 | 47.69 | 59.27 | 23.99 | 25.62 | 17.40 | 64.91 | 52.96 | 23.87 | 42.10 | 0.47 | 6.92 | 32.43 |
| | 100B×3 SP InvSqrt | 43.20 | 20.31 | 51.35 | 60.70 | 21.91 | 25.38 | 18.20 | 68.34 | 52.01 | 19.71 | 35.91 | 1.94 | 14.35 | 33.33 |
| | 100B×3 SP CosineInf | 43.22 | 22.61 | 51.05 | 59.22 | 22.78 | 26.48 | 18.60 | 68.17 | 53.51 | 23.01 | 36.90 | 1.83 | 14.29 | 34.00 |
| | 100B×3 SP Cosine | 43.38 | 20.05 | 50.46 | 60.06 | 22.75 | 25.23 | 18.00 | 67.57 | 52.17 | 23.13 | 39.74 | 1.44 | 13.40 | 33.65 |

TfQA: Truthful QA, WG:WinoGrande, NQ: Natural Questions, OBQA: OpenBook QA, TrQA:TriviaQA

Table 11: **All Zero-shot and Few-shot results on popular German LM benchmarks.** Normalized accuracy is reported for HellaSwag. All other tasks report unnormalized accuracy. MMLU is evaluated 5-shot, while all other tasks are zero-shot. Unexpectedly, we observe **on average**, that English language models match the performance of their German counterparts. Further inspection shows that performance is close to random chance on MMLU and ARC-C for all models, while German models perform better on HellaSwag and English models perform better on Truthful QA. This has the effect of canceling out the perceived improvements in the overall score.

| Training Tokens | MMLU DE | ARC-C DE | HellaSwag DE | TruthfulQA MC1 DE | AVG |
|---|---|---|---|---|---|
| 300B Pile | 24.92 | 18.77 | 27.09 | 26.93 | 24.43 |
| 300B SP | 23.28 | 17.49 | 27.03 | 27.91 | 23.93 |
| 300B Pile → 300B SP | 25.34 | 17.32 | 27.43 | 26.81 | 24.22 |
| 300B Pile → 300B SP (0.5% Replay) | 25.95 | 17.83 | 27.35 | 24.72 | 23.96 |
| 300B Pile → 300B SP (1% Replay) | 25.44 | 19.03 | 27.62 | 25.34 | 24.36 |
| 300B Pile → 300B SP (5% Replay) | 24.82 | 16.89 | 27.09 | 25.95 | 23.69 |
| 300B Pile → 300B SP (10% Replay) | 25.24 | 19.11 | 27.39 | 26.68 | 24.61 |
| 300B Pile → 300B SP (50% Replay) | 25.11 | 18.86 | 27.51 | 27.17 | 24.66 |
| 600B Pile ∪ SP | 24.43 | 18.26 | 27.36 | 26.44 | 24.12 |
| 200B Ger. | 23.74 | 18.26 | 29.53 | 26.32 | 24.46 |
| 300B Pile → 200B Ger. | 24.29 | 19.62 | 31.23 | 24.85 | 25.00 |
| 300B Pile → 200B Ger. (1% Replay) | 23.62 | 19.62 | 31.21 | 24.72 | 24.80 |
| 300B Pile → 200B Ger. (5% Replay) | 24.82 | 18.94 | 31.03 | 26.68 | 25.37 |
| 300B Pile → 200B Ger. (10% Replay) | 23.51 | 20.05 | 31.21 | 25.58 | 25.09 |
| 300B Pile → 200B Ger. (25% Replay) | 23.78 | 19.20 | 31.04 | 25.58 | 24.90 |
| 300B Pile → 200B Ger. (50% Replay) | 23.80 | 20.48 | 30.91 | 24.60 | 24.95 |
| 500B Pile ∪ Ger. | 24.53 | 18.43 | 30.45 | 25.70 | 24.78 |

### A.7 Aggregated average final accuracy

Table 12: **Final loss of models reported in sections 6.1, 6.2, 6.3, 6.4, and 7.4.** The loss is averaged over the last 100 iterations of training sampled at intervals of 10 iterations. The standard error is $\leq 0.001$ for all models so we don't report the specific value. We note, however, that this averaging does not correspond to Monte Carlo sampling over different random seems and is merely meant to reduce noise. We observe that models using more replay achieve a better adaptation-forgetting trade-off (AVG Loss). Interestingly, the model using 50% replay archives nearly identical loss values while seeing 150B fewer tokens on SlimPajama. All evaluation

| Model Size | Training Tokens | Max LR | Schedule | Validation Loss | | |
|---|---|---|---|---|---|---|
| | | | | $\mathcal{D}_0$ (Pile) | $\mathcal{D}_0$ (SP/German) | AVG |
| | 300B Pile → 300B SP | Constant | Cosine | 2.42 | 2.55 | 2.48 |
| | | $1.5 \times 10^{-4}$ | Cosine | 2.43 | 2.51 | 2.47 |
| | | $3 \times 10^{-4}$ | Cosine | 2.44 | 2.50 | 2.47 |
| | | $6 \times 10^{-4}$ | Cosine | 2.48 | 2.50 | 2.49 |
| | 300B Pile → 200B Ger. | Constant | Cosine | 3.22 | 1.21 | 2.22 |
| | | $1.5 \times 10^{-4}$ | Cosine | 3.47 | 1.13 | 2.30 |
| | | $3 \times 10^{-4}$ | Cosine | 3.56 | 1.11 | 2.34 |
| | | $6 \times 10^{-4}$ | Cosine | 3.63 | 1.11 | 2.37 |
| | 300B Pile | $3 \times 10^{-4}$ | Cosine | 2.17 | 2.70 | 2.44 |
| | 300B SP | $3 \times 10^{-4}$ | Cosine | 2.51 | 2.53 | 2.52 |
| | 200B Ger. | $3 \times 10^{-4}$ | Cosine | 3.97 | 1.17 | 2.57 |
| | 500B Pile ∪ 200B Ger. | $3 \times 10^{-4}$ | Cosine | 2.26 | 1.25 | 1.75 |
| | 300B Pile ∪ 300B SP | $3 \times 10^{-4}$ | Cosine | 2.17 | 2.53 | 2.35 |
| 405M | 300B Pile → 300B SP | $3 \times 10^{-4}$ | Cosine | 2.44 | 2.50 | 2.47 |
| | 300B Pile → 300B SP (0.5% Replay) | $3 \times 10^{-4}$ | Cosine | 2.27 | 2.50 | 2.39 |
| | 300B Pile → 300B SP (1% Replay) | $3 \times 10^{-4}$ | Cosine | 2.26 | 2.50 | 2.38 |
| | 300B Pile → 300B SP (5% Replay) | $3 \times 10^{-4}$ | Cosine | 2.23 | 2.51 | 2.37 |
| | 300B Pile → 300B SP (10% Replay) | $3 \times 10^{-4}$ | Cosine | 2.21 | 2.51 | 2.36 |
| | 300B Pile → 300B SP (50% Replay) | $3 \times 10^{-4}$ | Cosine | 2.16 | 2.54 | 2.35 |
| | 300B Pile → 200B Ger. | $3 \times 10^{-4}$ | Cosine | 3.56 | 1.11 | 2.34 |
| | 300B Pile → 200B Ger (1% Replay) | $3 \times 10^{-4}$ | Cosine | 2.83 | 1.12 | 1.97 |
| | 300B Pile → 200B Ger (5% Replay) | $3 \times 10^{-4}$ | Cosine | 2.57 | 1.12 | 1.85 |
| | 300B Pile → 200B Ger (10% Replay) | $3 \times 10^{-4}$ | Cosine | 2.46 | 1.13 | 1.80 |
| | 300B Pile → 200B Ger (25% Replay) | $3 \times 10^{-4}$ | Cosine | 2.33 | 1.16 | 1.75 |
| | 300B Pile → 200B Ger (50% Replay) | $3 \times 10^{-4}$ | Cosine | 2.24 | 1.22 | 1.73 |
| 10B | 300B Pile → 300B SP | $1.2 \times 10^{-4}$ | Cosine | 1.98 | 2.00 | 1.99 |
| | 300B SP | $1.2 \times 10^{-4}$ | Cosine | 2.08 | 2.05 | 2.07 |
| | 300B Pile | $1.2 \times 10^{-4}$ | Cosine | 1.75 | 2.24 | 1.99 |
| | 600B Pile ∪ SP | $1.2 \times 10^{-4}$ | Cosine | 1.72 | 2.02 | 1.87 |
| | 300B Pile → 300B SP (5% Replay) | $1.2 \times 10^{-4}$ | Cosine | 1.79 | 2.00 | 1.89 |
| 405M | 300B SP | $3 \times 10^{-4}$ | Cosine Inf | 2.51 | 2.53 | 2.52 |
| | | | InvSqrt Inf | 2.51 | 2.54 | 2.53 |
| | | | Cosine | 2.51 | 2.53 | 2.52 |
| | 100B×3 SP | $3 \times 10^{-4}$ | Cosine Repeats | 2.53 | 2.55 | 2.54 |
| | | | Cosine Inf | 2.58 | 2.61 | 2.59 |
| | | | Cosine | 2.59 | 2.61 | 2.60 |

## B Model hyperparameters

| Description | Value |
|---|---|
| **10B Transformer Model** | |
| Parameters | $9,642,249,216$ |
| Non-Embedding Parameters | $9,408,678,912$ |
| Num layers | 36 |
| Hidden size | 4608 |
| Num attention heads | 36 |
| **405M Transformer Model** | |
| parameters | $405,334,016$ |
| Non-Embedding Parameters | $353,822,720$ |
| Num layers | 24 |
| Hidden size | 1024 |
| Num attention heads | 16 |
| **Common** | |
| Optimizer | AdamW |
| $\beta_1,\beta_2$ | $0.9, 0.95$ |
| Batch size | 1104 |
| Sequence length | 2048 |
| Hidden activation | GeLU |
| Weight decay | 0.1 |
| Gradient clipping | 1.0 |
| Decay | Cosine |
| Positional embedding | Rotary |
| GPT-J-Residual | True |
| Weight tying | False |
| Vocab Size | 50432 |
| Rotary PCT | 0.25 |

Table 13: **Hyperparameters of our 405M and 10B parameter transformer LLMs.**

| Description | Value |
|---|---|
| **10B Model- Cosine Schedule** | |
| Max learning rate ($\eta_{max}$) | $1.2 \cdot 10^{-4}$ |
| Min learning rate ($\eta_{min}$) | $1.2 \cdot 10^{-5}$ |
| Warmup percent ($T_{warmup}$) | 1 |
| **405M Model - Cosine Schedule** | |
| Max learning rate ($\eta_{max}$) | $3 \cdot 10^{-4}$ |
| Min learning rate ($\eta_{min}$) | $3 \cdot 10^{-5}$ |
| Warmup percent ($T_{warmup}$) | 1 |
| **405M Model - Infinite LR Schedule Common** | |
| Max learning rate ($\eta_{max}$) | $3 \cdot 10^{-4}$ |
| Min learning rate ($\eta_{min}$) | $3 \cdot 10^{-5}$ |
| Constant learning rate ($\eta_{const}$) | $1.65 \cdot 10^{-4}$ |
| Warmup percent ($T_{warmup}$) | 1 |
| Cooldown iters percent ($T_{cd}$) | 60 |
| Constant iters percent ($T_{ann}$) | 25 |
| **Inverse Square root cooldown schedule** | |
| Timescale ($\alpha$) | 10 |

Table 14: **Hyperparameters of LR schedules.** Unless otherwise specified in the text, we use these values.

