# Continual Pre-Training of Large Language Models: How to (re)warm your model?

## Abstract

Large language models (LLMs) are routinely pre-trained on billions of tokens, only to restart the process over again once new data becomes available. A much cheaper and more efficient solution would be to enable the continual pre-training of these models, i.e. updating pre-trained models with new data instead of re-training them from scratch. However, the distribution shift induced by novel data typically results in degraded performance on past data. Taking a step towards efficient continual pre-training, in this work, we examine the effect of different warm-up strategies. Our hypothesis is that the learning rate must be re-increased to improve compute efficiency when training on a new dataset. We study the warmup phase of models pre-trained on the Pile (upstream data, 300B tokens) as we continue to pre-train on SlimPajama (downstream data, 297B tokens), following a linear warmup and cosine decay schedule. We conduct all experiments on the Pythia 410M language model architecture and evaluate performance through validation perplexity. We experiment with different pre-training checkpoints, various maximum learning rates, and various warmup lengths. Our results show that while rewarming models first increases the loss on upstream and downstream data, in the longer run it improves the downstream performance, outperforming models trained from scratch—even for a large downstream dataset.

## 1. Introduction

Large pre-trained models have enabled massive performance improvements for many downstream tasks in vision (Kirillov et al., 2023; Oquab et al., 2023) and language (Brown et al., 2020; Zhao et al., 2023). However, training these foundation models is prohibitively expensive. Existing works aim to reduce the cost of large-scale model development by enabling low-cost hyperparameter optimization (Yang et al., 2022) or providing guidelines for maximizing performance under a given compute budget (Hoffmann et al., 2022). However, these works assume that models will be *trained from scratch*. As the amount of data available for pre-training is ever-growing, new and improved datasets (e.g. RedPajama and SlimPajama (Together.xyz, 2023; Soboleva et al., 2023; Touvron et al., 2023)) will continue to become available. Should practitioners always combine existing datasets (e.g. Pile (Gao et al., 2020)) and *train from scratch* to obtain the best performance? Doing so would quickly become prohibitively expensive and fails to leverage existing pre-trained models.

Our approach circumvents the need for complete re-training by continuing to pre-train existing models on new data. We refer to this as "continual pre-training" and the goal is to minimize the loss on new data while maintaining low loss on previous data. Continual pre-training is a critical challenge since it can lead to catastrophic forgetting (French, 1999). Moreover, the potential long sequence of training stages may make common continual learning techniques such as replay (Rebuffi et al., 2017; Ostapenko et al., 2022) or regularisation (Kirkpatrick et al., 2017; Farajtabar et al., 2020) not compute efficient enough (Lesort et al., 2023). A simple and – from a compute cost perspective – scalable solution to limit forgetting in such situations is to (only) progressively decrease the learning rate every time new data becomes available (Mirzadeh et al., 2020; Winata et al., 2023). However, this solution is limited because repeatedly decreasing the learning rate would cause it to eventually become too small if the number of training stages becomes high.

In this work, we take a step towards efficient continual pre-training by studying how to re-increase a small learning rate to keep training a pre-trained language model on new data. We refer to this as *re-warming* the model. Re-warming the model should improve learning efficiency by avoiding a vanishing learning rate. We study warm-up strategies on Pythia 410M model with various amounts of data, maximum learning rates and different pre-trained checkpoints. This

[1]Anonymous Institution, Anonymous City, Anonymous Region, Anonymous Country. Correspondence to: Anonymous Author <anon.email@domain.com>.

Preliminary work. Under review by the International Conference on Machine Learning (ICML). Do not distribute.

would allow a model trained initially on a large dataset to benefit from resuming training on a newer large dataset without having to retrain from scratch. In order to simulate this setting, we fix our initial pre-training dataset to be Pile and the newer dataset to be SlimPajama. We hope that this may guide the adaptation of existing LLMs to future new datasets.

Our results show that:

1. Progressively increasing the learning rate to warm-up is not necessary but starting directly from the maximum learning rate creates an initial large spike in the loss (chaotic phase a.k.a stability gap) with no consequences later.
2. Adjusting the maximum learning rate can help trade-off between upstream and downstream performance; increasing the maximum learning rate leads to stronger adaptation to the downstream dataset (SlimPajama), while smaller learning rates preserve more performance on the upstream dataset (Pile).
3. Continual pre-training with the latest pre-trained checkpoint improves performance.

## 2. Setup

In our setup, the upstream (or pre-training) dataset is the Pile (Gao et al., 2020). The downstream (or fine-tuning) dataset is SlimPajama (Soboleva et al., 2023). SlimPajama is an extensively deduplicated version of RedPajama (Together.xyz, 2023) which is built based on the LLama dataset (Touvron et al., 2023). In this work, we use "fine-tuning" and downstream continual pre-training interchangeably. However, in our continual pre-training setting, we note that the downstream dataset is on the scale of the previous pre-training dataset (i.e. very large, unlike many fine-tuning datasets).

The SlimPajama dataset is built from similar sources as the Pile but with a higher quantity of data. Therefore, some upstream data may be repeated during downstream pre-training. Our experimental setup is comparable to the setup of (Ash & Adams, 2020), where they train a classifier on half of the samples of a dataset first, and fine-tune it later on all samples. They show that warm starting for image classification is challenging. Using a model pre-trained on the Pile and continuing the pre-training on SlimPajama, we follow an analogous setup for causal language modeling.

**Datasets –** We use the Pile with the same weights as Black et al. (2022) for validation. We shuffle and randomly sample the SlimPajama dataset (Soboleva et al., 2023) to form the ∼297B token training dataset and ∼316M validation token dataset. We do not use replay. We use the same tokenizer as (Black et al., 2022) that is trained specifically on the Pile.

**Model –** We use the 410M Pythia pre-trained on the Pile

*Table 1.* Token counts and train data weights for our subsampled version of SlimPajama.

| Dataset | Sampling % | Train | Val |
|---|---|---|---|
| StackExchange | 2.0 | 9.95B | 13.08M |
| Arxiv | 2.5 | 13.77B | 22.73M |
| Wikipedia | 4.5 | 11.78B | 15.79M |
| Book | 4.5 | 14.22B | 22.04M |
| Github | 4.5 | 15.41B | 22.42M |
| C4 | 15.0 | 78.49B | 72.49M |
| Commoncrawl | 67.0 | 153.25B | 147.28M |
| Totals | 100 | 296.86B | 315.83M |

(Biderman et al., 2023), i.e. GPT-NeoX (Black et al., 2022) models. We do not use flash attention (Dao et al., 2022).

**Hyperparameters –** We use the AdamW optimizer with $\beta_1 = 0.9, \beta_2 = 0.95, \epsilon = 10^{-8}$, and a weight decay of $0.1$. The maximum learning rate is varied in our experiments $\{1.5 \cdot 10^{-4}, 3 \cdot 10^{-4}, 6 \cdot 10^{-4}\}$. We use cosine learning rate decay to a minimum of $0.1 \cdot MaxLr$. All warmup lengths are calculated based on the full downstream dataset size (297B tokens). We note that our cosine decay schedule reaches the minimum learning rate at 240B tokens and is constant thereafter. We set gradient clipping to $1.0$. Training is conducted at half-precision (FP16), without dropout.

## 3. Related Work

**Large Language Models:** LLMs are usually trained with Adam (e.g., GPT3 (Brown et al., 2020), BLOOM (Scao et al., 2022), Gopher (Rae et al., 2021), Pythia (Biderman et al., 2023)) or AdamW (e.g., Chinchilla (Hoffmann et al., 2022), LLaMA (Touvron et al., 2023)). In all the aforementioned models, the learning rate schedule consists of a warm-up followed by a cosine decay to 10% of the maximum learning rate.

**Unsupervised Continual Learning:** In this paper, we investigate various warm-up strategies for the continual pre-training of LLMs. Continual pre-training uses a similar type of training objectives as continual self-supervised training. Self-supervised pre-training was also studied in vision datasets for image generation (Seff et al., 2017; Lesort et al., 2019; Zhai et al., 2019; Nguyen et al., 2018; Davari et al., 2022) or representation learning (Fini et al., 2022; Madaan et al., 2021; Rao et al., 2019). In language, continual pre-training was studied under the name of domain adaptation pre-training (Ke et al., 2023a; Scialom et al., 2022; Gururangan et al., 2021; Qin et al., 2022) where the new dataset comes from a new domain. Another setting is where different datasets are generated at different points in time (Han et al., 2021; Jin et al., 2022; Jang et al., 2021; 2022; Loureiro

et al., 2022). In our setup, the scenario is closer to domain adaptation pre-training, because we do not take into account the temporality of data.

**Monitoring Learning Rate for Continual Training of Language Models:** In continual learning (CL), models are trained on sequences of datasets. Therefore, the data is not *independent and identically distributed* which can lead the model to lose plasticity or forget. In such situations, particular monitoring of the learning rate schedule can be beneficial. In CL of language models (Caccia et al., 2021; Ke et al., 2023a; Loureiro et al., 2022; Han et al., 2021; Loshchilov & Hutter, 2018; Scialom et al., 2022; Winata et al., 2023) different approaches have been evaluated: constant learning rate (Ke et al., 2023a; Scialom et al., 2022), progressive decrease (Winata et al., 2023) or warm-up then decrease (Caccia et al., 2021).

However, to the best of our knowledge, no existing work studies specifically the influence of the warm-up phase in the context of continual pre-training for large language models.

## 4. Continual Warm-up

### 4.1. How long to warm up?

In the literature, warm-up is usually conducted on at most 1% of the data (Zhao et al., 2023). In this experiment, we investigate if the results are sensitive to this hyper-parameter.

**Setup:** We experiment with different warm-up lengths for a schedule of 297B tokens: 0%, 0.5%, 1%, and 2% of the data and measure the performance after the first 50B tokens. From a different perspective, we could see this experiment as running a 1% warm-up on different amounts of data. We hypothesize that warming up for a larger number of iterations could lead to a smoother transition with subsequent performance improvements.

**Results:** The results of this experiment are provided in Fig. 1. They show that the amount of data used for warming up the learning rate does not significantly influence the perplexity on the downstream task (learning) or the upstream task (forgetting). These results invalidate our hypothesis that using more tokens for warm-up can smooth the transition and show that linear warmup is useless in this setting. Nevertheless, the model trained without any progressive warm up experiences an initial "choatic phase" causing a spike in the loss in its first few iterations of training, this phenomenon is also referred to as stability gap (Lange et al., 2023; Caccia et al., 2022).

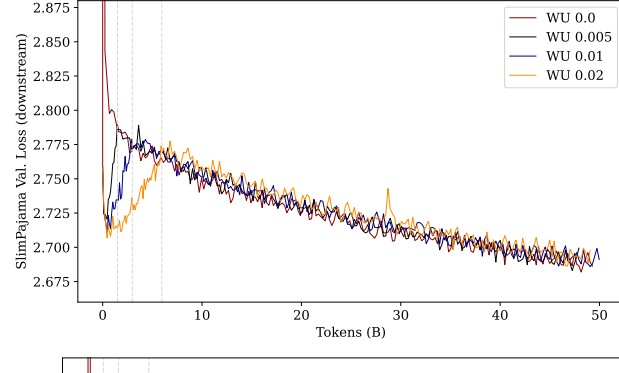

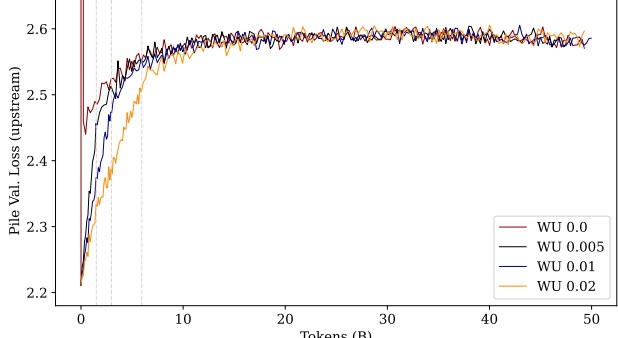

Figure 1. (*top*) Evolution of perplexity on SlimPajama while fine-tuning with various amounts of tokens for warm-up. (*bottom*) perplexity on the same experiments on the Pile validation set (upstream). $MaxLr = 3 \cdot 10^{-4}$, $MinLr = 0.1 \cdot MaxLr$. This figure shows that at that scale, the length of the warm-up phase does not significantly influence results.

> **Takeaway 1:**
>
> - The length of the warmup phase does not appear to have a significant effect on the Pile and SlimPajama validation losses.

### 4.2. How high to warm up?

One objective of re-warming the learning rate is to enable compute-efficient continual pre-training. A learning rate that is too small may lead to inefficient learning on the downstream dataset, whereas, a learning rate that is too large may lead to catastrophic forgetting of the upstream dataset. One important aspect of re-warming the learning rate is to decide how high to increase it. Therefore, in this experiment, we vary the maximum learning rate to assess its effect on performance.

**Setup:** We fix the length of the warm-up phase to the default amount of 1% of the training data and vary the maximum learning rate. We experiment with the default value of $3 \cdot 10^{-4}$ used for pre-training Pythia 410M (Biderman et al., 2023), $1.5 \cdot 10^{-4}$, and $6 \cdot 10^{-4}$. For the post-warmup cosine decay phase, we set the final learning rate to 10% of the

maximum learning rate. The learning rate schedule we used decays to the minimum learning rate at 240B tokens and is constant thereafter. The runs are reported to the end of 240B tokens (the end of decay period).

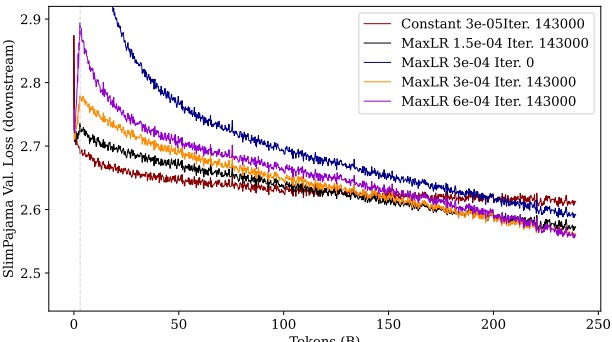

*Figure 2.* Evolution of loss on SlimPajama for different maximum learning rates. The blue curve reports a model trained from scratch. Growing the maximum learning rate consistently decreases the final loss on downstream data. At convergence, the models being continually pre-trained outperform the scratch and constant LR baselines. However, the constant learning rate model achieves best performance within the first 100B tokens.

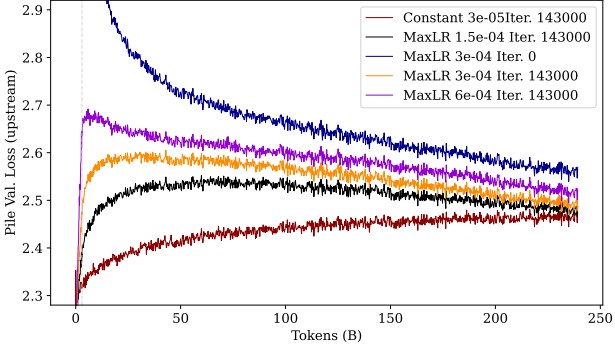

*Figure 3.* Evolution of loss on Pile for different maximum learning rates. The blue curve reports a model trained from scratch. Growing the maximum learning rate consistently increases the final loss on upstream data, i.e. it increases forgetting. The from-scratch baseline consistently improves its performance on Pile, while being trained on SlimPajama, showing the significant synergy between both datasets.

**Results:** The results of this experiment are provided in figures 2, 3, and 4. We observe, at the end of training, that larger maximum learning rates improve performance on downstream data, while they hurt performance on upstream data. Conversely, a smaller maximum learning rate improves performance on upstream data, while limiting adaptation to downstream data—causing decreased performance. These findings show that altering the maximum learning rate can be an effective way to tradeoff between downstream and upstream performance. Additionally, we observe a gen-

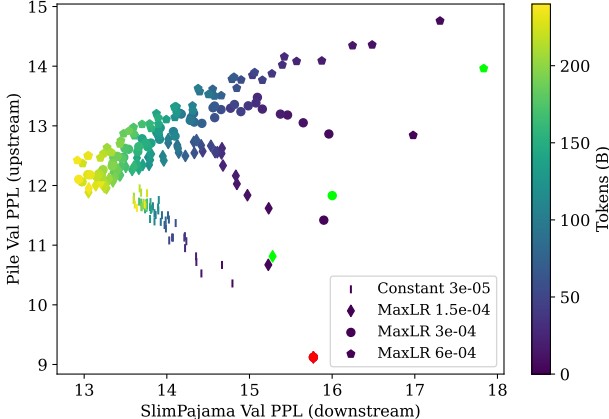

*Figure 4.* Perplexity downstream vs perplexity upstream, RP fine-tuning. Green points refer to the ends of the warm-up phases. The red point represents the perplexity before starting the downstream fine-tuning. Increasing the maximum learning rate improves performance on the downstream data, but causes forgetting on the upstream. This plot reports the same results as figures 2 and 3.

eral trend: fine-tuning on SlimPajama, causes the model to forget what has been learned on the Pile leading to an increase in the Pile validation perplexity. Finally, we note that employing early stopping on the model trained from a constant learning rate (similar to traditional fine-tuning) is an economical way of adapting to the new data distribution while retaining strong performance on the upstream dataset.

> **Takeaway 2:**
>
> - Rewarming then decaying the learning rate appears necessary to learn well on the downstream task. Moreover, while keeping a constant learning is initially advantageous on Pile, this advantage vanishes when training long enough on SlimPajama.
>
> - A model that only learns on SlimPajama performs worse on SlimPajama than models pretrained on Pile in spite of being optimised solely for the downstream task, highlighting positive transfer between the two datasets.

### 4.3. Comparing with from Scratch Training

In this experiment, we want to compare finetuned models with models trained from scratch.

**Setup:** We train a model from random initialization using the same cosine decay schedule as the $MaxLr = 3 \cdot 10^{-4}$ model in Section 4.2.

**Results:** As we can see in Fig. 2 and Fig. 3, all the finetuned models with a warm-up perform better than the model

trained from scratch. This shows that finetuning instead of retraining might improve performance even when the downstream dataset is on the scale of the upstream dataset and overlaps with the upstream dataset. We also observe that, after 200B tokens, the model trained from scratch performs better than the model finetuned using a constant learning rate.

### 4.4. Re-warming on the same data

In the previous experiments, we have seen that finetuning on new data leads to a quick increase of loss on past data, that decreases later. The increase is higher when the max learning rate is bigger. One hypothesis for the increase in loss is that the distribution shift between upstream and downstream data disturbs the training process. To assess this hypothesis, we apply our warm-up policy in a setting with no distribution shift. That is, we replicate our experiments from figures 3 and 4 by fine-tuning on Pile.

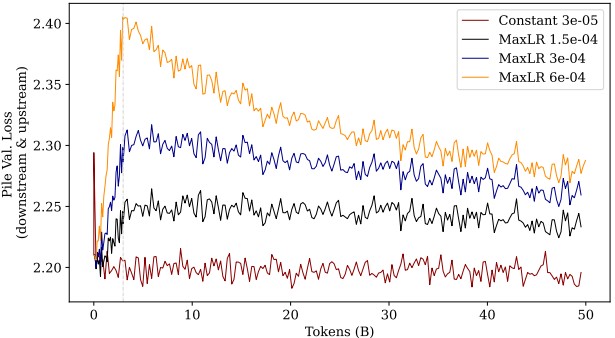

*Figure 5.* Pile validation loss while fine-tuning again on the Pile. Warm-up phenomenon observed in Sec. 4.2 is also observed applied to fine-tuning again on the same data distribution. Warm-up token=1% downstream tokens, $MinLr = 0.1 \cdot MaxLr$.

**Setup:** In this experiment, instead of fine-tuning on SlimPajama data, we fine-tune on 50B tokens of the Pile data with the same parametrization of the warm-up policy as Sec. 4.2 experiments.

**Results:** Fig. 5, shows that re-warming the learning rate while continuing to pre-train on the Pile has a similar effect as re-warming on SlimPajama data Fig. 3 when looking at the downstream validation loss. This suggests that the distribution shift between Pile and SlimPajama is not solely to blame for the negative impact of re-warming the learning rate observed in sec. 4.2, and that the optimization dynamics also plays a role in this increase of loss.

Fig. 6 shows that the training first increases perplexity on both the Pile and SlimPajama data but reduces after on both. Interestingly, Fig. 6 show a linear relationship between SlimPajama perplexity and the Pile perplexity when fine-tuning on the Pile, while it was not the case while fine-

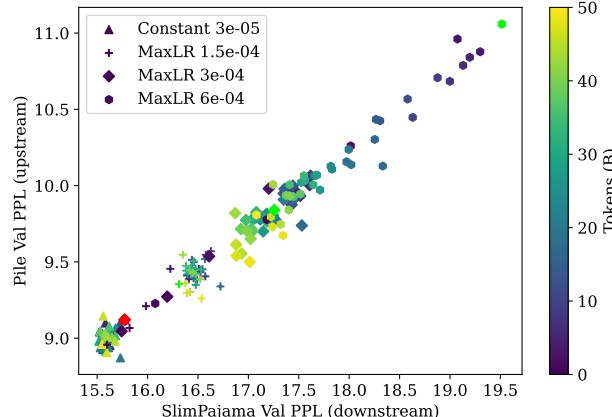

*Figure 6.* Perplexity on the Pile vs perplexity on SlimPajama when fine-tuning on the Pile with various maximum learning rates. Warm-up token=1% downstream tokens, $MinLr = 0.1 \cdot MaxLr$. Green points refer to the end of the warm-up phase.

tuning on SlimPajama (Fig. 3). One possible explanation for this relationship is that models trained on Pile climb out of a minimum during warmup and return towards the same minimum as the learning rate is decayed, yielding the linear trend.

---

**Takeaway 3:**

- Rewarming the learning rate appears to be a significant cause for the degradation of performance seen previously when starting to learn on the downstream task, as evidenced by rewarming then decaying the learning rate while training on the same dataset.

- The models struggle to recover from the performance hit due to rewarming the learning rate when training on the same dataset.

---

### 4.5. Evaluating Earlier Checkpoints

**Setup:** We select three checkpoints from model pre-training to test if warm-up strategies benefit from starting with non-converged checkpoints. Our hypothesis is that selecting checkpoints farther from convergence may benefit adaptation to the downstream task as these checkpoints may be located at more favorable points in the loss landscape.

To select significantly different checkpoints, we compare the last pre-training checkpoint (i.e. Pythia 410M after $143,000$ iters), to an earlier checkpoint achieving a Pile validation loss near the maximum Pile validation loss attained by all models in Fig. 1 (bottom) ($\sim 2.5$), and a third checkpoint in between the two other checkpoints.

**Results:** The evolution of the validation losses on SlimPa-

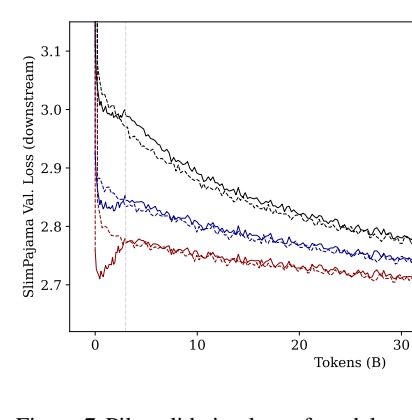

*Figure 7.* Pile validation loss of models trained from the fully converged checkpoint (red), the upstream saturation point (black), and 1/2 of the upstream saturation point (blue).

jama are provided in Fig. 7 and the evolution of the validation losses on the Pile is provided in appendix A. We see in Fig. 7 that, in our setup, selecting earlier checkpoints for later fine-tuning does not lead to improvement in downstream performance. Therefore, selecting the latest checkpoint is the best option. We can conclude that the pre-training did not lead the model into a loss of plasticity that would make the model difficult to re-warm.

**Local conclusion:** The experiments conducted in this section led to the conclusion that re-warming the pre-trained model on new data is a challenging task, even when the downstream data is of similar provenance to the upstream data. Our results show that the amount of tokens used for warm-up does not significantly alter performance, growing the maximum learning rate improves downstream performance of the final model while decreasing it improves upstream performance, and selecting earlier checkpoints decreases performance on both upstream and downstream data.

> **Takeaway 4:**
>
> • Using an earlier checkpoint of Pile pretraining does not lead to learning faster on SlimPajama.

## 5. Discussion / Limitation

**Data similarity and overlapping:** In our experimental setup, upstream and downstream data have a high similarity, notably because of data overlap. Since in continual learning, different types of shifts can lead to variations in performance (Lesort et al., 2021), our results may not generalize to setups with different distribution shifts, such as language domain adaptation pre-training setups (Xu et al., 2019; Gururangan et al., 2020; Ke et al., 2023a; Chakrabarty et al., 2019; Ke et al., 2023b). Nevertheless, comparing Fig. 4 and Fig. 6,

we see that the results are not identical when fine-tuning on the Pile or when fine-tuning on SlimPajama. A possible explanation is that even a slight shift in data distribution can lead to a significant perturbation of the learning dynamics. For example, in the context of image classification, Igl et al. (2020) show how a sudden transition of 10 to 20 % of the labels in the dataset can have a significant impact on the downstream performance (see Fig. 5 of (Igl et al., 2020)).

**Experiments Scale:**

As described in Sec. 2, our investigation explores models of size 410M and fine-tuning dataset of size 297B tokens. While this is a preliminary study, in future work, we plan to verify whether our conclusions hold at different model scales (e.g., 3B and 7B) and different dataset scales (e.g., 100B and 600B). Moreover, we plan to test our models throughout using benchmarks such as HELM (Liang et al., 2022) or Harness (Gao et al., 2021) instead of only loss or perplexity, as these benchmarks can provide important insight into the evolution of model capabilities.

## 6. Conclusion

Our experiments demonstrate that warming up to higher maximum learning rates helps models pre-trained on the Pile adapt to SlimPajama, while a smaller maximum learning rater preserves performance on the pile. In both cases, however, models that are rewarmed improve over models trained from scratch. These results motivate the use of continual pre-training on new datasets rather than restarting training from scratch. More research is needed, however, to establish similar results for larger model scales, different distribution shifts, and verify that this strategy can be applied repeatedly to update models.

## Software and Data

GPT-NeoX (Andonian et al., 2021), DeepSpeed (Rasley et al., 2020), nccl (NVIDIA, 2016), Apex (NVIDIA, 2019), Pytorch (Paszke et al., 2017), HuggingFace Transformers library (Wolf et al., 2020).

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

# A. Upstream loss when fine-tuning various checkpoints.

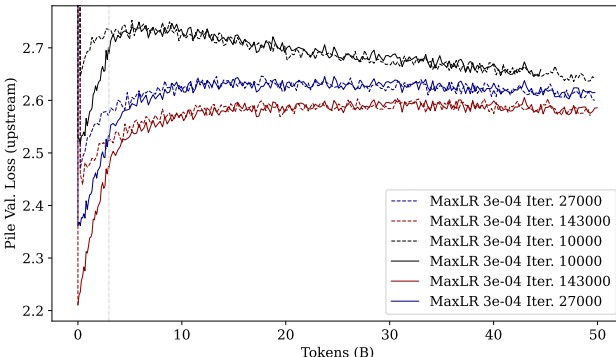

*Figure 8.* Pile validation loss of models trained from the fully converged checkpoint, the upstream saturation point, and $1/2$ of the upstream saturation point. The experiments for this figure are described in Sec. 4.5.

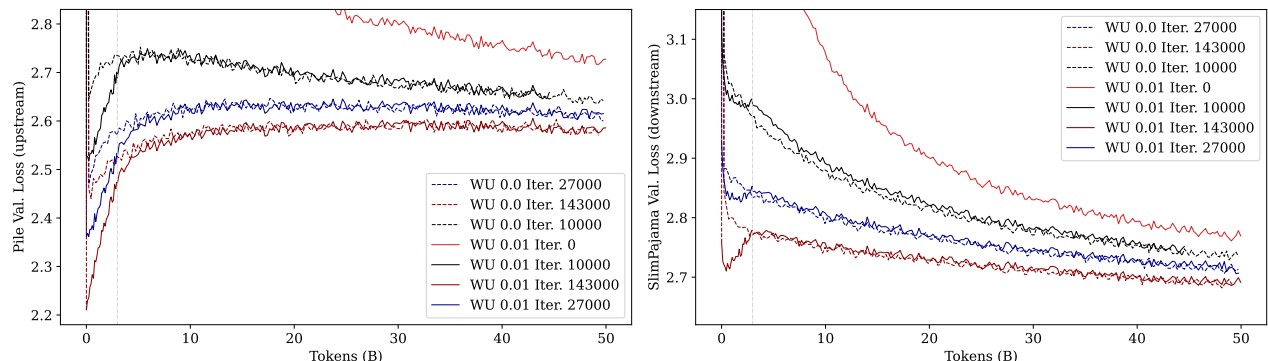

*Figure 9.* Training from a pre-trained checkpoint achieves lower Pile and SlimPajama validation loss faster than training from scratch.