# OpenReview forum: "Simple and Scalable Strategies to Continually Pre-train Large Language Models"
_TMLR — Accepted by TMLR_

### Review · Reviewer_663b · 2024-03-28

**Summary Of Contributions:**

This paper studies the best way to continually pretrained LLMs. They argue that rewarming and redecaying the LR and replaying samples can help LLMs learn the new pre-training mixture w/o forgetting the old mixture. The main setup is the Pile to SlimPajama and the Pile to the German Common Crawl.

**Audience:**

Yes

**Claims And Evidence:**

Yes

**Requested Changes:**

Major
- Though clearly stated the experimental setup, the conclusions should reflect the experimental setup and be more limited. For example,
1.  Intro: “To answer this question, we conduct a large-scale empirical study of continual learning techniques for LLM pre-training.” The empirical study was for GPT-NeoX pre-training specifically
2. Intro - “We establish the effect of learning rate re-warming and re-decaying for models pre-trained using a cosine schedule”. The effect was only shows for GPT-NeoX pre-trained.

Minor
- A graph showing the LR vs timestep for cosine decay schedule would help the reader more easily understand what the scheduler is doing
- “ therefore likely to have newer data within the overlapping domains” → What exact data sources are in SlimPajama and not the Pile?
- How did you choose the subsampling percentages for the different datasets in SlimPajama?
- In Figure 2a and 2c, why is there an initial spike in the validation loss for 0% linear warmup?
- In figure 2b, why is there an increase in the validation loss between 2B to 10B tokens for 0.5%, 1%, and 2% linear warmup?
- In 6.1.2, what about a baseline that starts at n_max and re-decays without rewarming (i.e. it uses the same scheduler as pre-training)
- In re-warming and re-decaying, is the model re-warmed the first half and re-decayed the second half?

Typos
- 5.1 “and validaiton sets”  → validation
- 5.3 - “adamW” → “AdamW”
- 5.3 - “ batch size of 1104” → 1024?
- “GPU nodes” → What type of GPU nodes?
- 6.2 “However, when using an extreme amount of replay, … we observe that the model adapts less to D0” → Do you mean D1?

**Strengths And Weaknesses:**

Strengths
- Very relevant and useful question and practical solution proposed
- Thorough experimental setup

Weaknesses
- My biggest concern is how general the conclusions were, but how limited the setup was. I understand it takes a lot of compute to do the pre-training, as evident from the training setup described in section 5.3 However, I do not think that pre-training on just 2 setups with GPT-NeoX is enough evidence for the general conclusion.
- The takeaways from the results and graphs in section 6 were nicely stated, but it was not easy for me to follow all the graphs and results and how they justified the takeaway.

---

### Review · Reviewer_uTNo · 2024-04-18

**Summary Of Contributions:**

This paper proposes a scalable alternative solution to continuous pre-training of Large language models. The common problems for continuous pre-training are poor adapting of new data or forgetting the learned data. To overcome these challenges the authors have proposed a three-stage process- learning rate re-warming, learning rate re-decaying, and replay of previous data. The authors have also noted forgetting can still happen due to the learning rate re-warming and propose an infinite learning rate schedule.

**Audience:**

Yes

**Broader Impact Concerns:**

None.

**Claims And Evidence:**

Yes

**Requested Changes:**

1. In Appendix A6 - there is no mention of the 10B model performance on the infinite learning rate schedule. You have also mentioned this test was only performed on the 405M model. Can you explain the reason for that?

2. Can you please provide some details of compute consumption due to the replay rate change? Are those linearly related? From Figure 5b, it seems increasing the replay rate is lowering validation loss. So my question is - if changing the replay rate doesn't increase compute much- should we start with 50% replay than 5%?

3. The proposed step relies on replay. Can this be used in private data or if the data privacy need to be protected?

**Strengths And Weaknesses:**

The paper proposed an interesting idea in the continual pre-training step. The proposed 3 phase steps are simple and it will be good to see if the steps can be expanded on larger models or datasets. Also, the experiments are comprehensive and the authors did an excellent job of providing detailed results.

However, the result with the strong distribution shift dataset (German dataset) is lacking because there is no meaningful evidence. I understand that the authors have provided some possible explanations. The paper will be stronger if a different distributed shift dataset can be used where these results are more prominent.

Moreover, the addition of an infinite learning rate schedule proposal is only tested in a low-size model. There is no discussion about whether the learning rate pre-warm which is an essential step mentioned in this paper is scalable or whether the forgetting problem is going to be increased in comparison.

---

### Review · Reviewer_b7Jx · 2024-04-27

**Summary Of Contributions:**

This paper presents a method to more efficiently and effectively adapt the pretrained checkpoint to the new data by re-warming the learning rate scheduling to avoid performance degradation brought by the distribution shift.

**Audience:**

Yes

**Broader Impact Concerns:**

No broader impact concerns.

**Claims And Evidence:**

Yes

**Requested Changes:**

1. Can the author provide insights on how to choose the hyper-parameter like the replay rate to make the adapted model perform well on both original domain and the targeted domain, without running expensive hyper-parameter tuning?
2. Can you find a hyper-parameter that out-performs the ckpt trained on the union of the both datasets?
3. What if there're more new data coming, can this method be extended to >3 datasets?
4. Can the author provide more evaluation results using popular lm eval benchmarks, especially some more complex tasks like QA, commonsense reasoning, etc.? To show this is a general methods that suits modern models and more challenging benchmarks

**Strengths And Weaknesses:**

Strengths:
The paper presents a simple method and tested it on controlled and reasonably large scale dataset to show the effectiveness of this technique, which I really appreciate it. The method is clean and the paper is easy to follow. The technique could have high impact in the future if this holds effective in production scale and extend experiments to more distribution shift scenarios and show benefits in more challenging evaluation settings.

Weaknesses:
1. the learning rate scheduling rewarming or decaying has existed for a while as the paper indicated in Section 3.4, and the author claims their novelty to be setting up a controlled framework to verify this technique sounds incremental.
2. The re-decaying method doesn't preserve the loss on the original dataset that the checkpoint is trained on. Therefore, the adapted model is weaker than the model trained on the union of the both datasets. Can this be one reason why the industry prefers to train from scratch on the large new pool of data instead of initializing from one previous ckpt and skip that data source?
2. The replaying method looks good but the proposed method is sensitive to the hyper-parameter, i.e., randomly picking one hyper-parameter doesn't necessarily lead to better results than training on a union of the data.

---

> ### Author Response · Authors · 2024-05-03
> **General comments and replies to weaknesses**
>
> Thank you for your careful reading of our paper and thoughtful feedback. We are pleased that b7Jx finds our paper easy to follow and believes that our experiments show the effectiveness of our technique.
>
> > the learning rate scheduling rewarming or decaying has existed for a while as the paper indicated in Section 3.4, and the author claims their novelty to be setting up a controlled framework to verify this technique sounds incremental.
>
> As you have pointed out, sections 3 and 4 of our paper explain that replay and learning rate schedules have both been explored by previous work, albeit in different contexts. While some may argue that the combination and benchmarking of these methods is incremental, we believe that the relevance of our results to current trends in ML and the practicality of our simple strategies make our investigation worthwhile and potentially high impact (as you mentioned). Furthermore, rewarming and redecaying are not heavily studied in the Continual Learning (CL) literature as it has largely focused on smaller-scale image recognition settings where rewarming is not necessary.
>
> > The re-decaying method doesn't preserve the loss on the original dataset that the checkpoint is trained on. Therefore, the adapted model is weaker than the model trained on the union of the both datasets.
>
> As mentioned in section 6.1.2, re-decaying is needed for strong adaptation to the new dataset, however, models that re-decay “experience significant forgetting” on the original dataset, motivating the need for replay to make these models competitive. When using replay, as shown in Figure 1, continually pre-trained models can obtain similar average final validation and average evaluation performance to the union baseline.
>
>
> > Can this be one reason why the industry prefers to train from scratch on the large new pool of data instead of initializing from one previous ckpt and skip that data source?
>
> We believe that the industry decides to train from scratch due to the absence of evidence, until now, that continual pre-training is a viable alternative.

---

> ### Author Response · Authors · 2024-05-03
> **Reply to Requested Changes**
>
> > The replaying method looks good but the proposed method is sensitive to the hyper-parameter, i.e., randomly picking one hyper-parameter doesn't necessarily lead to better results than training on a union of the data.
>
> > Can the author provide insights on how to choose the hyper-parameter like the replay rate to make the adapted model perform well on both original domain and the targeted domain, without running expensive hyper-parameter tuning?
>
> We agree that picking the replay hyper-parameter at random may not lead to desirable results. In existing works (*Learning To Learn Without Forgetting By Maximizing Transfer And Minimizing Interference*. Reimer et al., *Dark Experience for General Continual Learning: a Strong, Simple Baseline*. Buzzega et al., etc…), tuning the replay-related hyperparameter(s) is standard practice. Predicting the best replay hyperparameter(s) value a-priori without any tuning is a considerable task and is out of the scope of this work. In our experiments (see figure 5), the replay trends can be estimated from a relatively small amount of training (e.g. enough to see the curves stabilize). Therefore, replay is relatively cheap to tune when continually pre-training LLMs. As part of our reply to uTNo, we edited the manuscript to highlight how one can easily tune this hyperparameter (e.g., see rules of thumb).
>
> > Can you find a hyper-parameter that out-performs the ckpt trained on the union of the both datasets?
>
> As stated in the introduction, “In this work, we do not intend to improve on the performance of models trained from a random initialization on all of the available data. Instead, we consider models trained on the union of existing datasets as baselines whose performance we seek to match using a combination of continual learning strategies at scale”.
>
> That being said, it is possible to select the replay hyperparameter that outperforms the checkpoint trained on the union of both datasets. In Figure 5, we observe, for the German transition, that the continually pre-trained model with 50% replay outperforms the union baseline on both datasets. For the English transition, the model trained with 50% replay matches the average validation performance of the union model. While such a high degree of replay comes at the cost of seeing less new data in our compute-equivalent replay setting (hindering the performance of the English model), expending more compute to ingest all new data, say with 50% replay, can be worthwhile in practice (e.g. increasing performance of the continually pre-trained English model on new data) and is still economical in many settings, even at industry scale. Take for instance LLama3 8B trained on 15T tokens, updating the model on 300B new tokens with 50% replay will necessitate training on 600B total tokens. This is 26x less training compared to re-training from scratch on 15.6T tokens.
>
> > What if there're more new data coming, can this method be extended to >3 datasets?
>
> Our results in Figure 10 show that it is possible to continue pre-training on new data using repeated cosine decay schedules (up to 3). It is reasonable to expect these trends to hold for more splits of Slim Pajama (3+), suggesting that the performance of continual pre-training (e.g., results from sections 6.3 and 6.4) can be extended without being negatively impacted by the schedule. However, we suspect there may be a need for more careful handling of the replay buffer (e.g. using techniques such as reservoir sampling). We leave explorations of these details to future work.
>
> > Can the author provide more evaluation results using popular lm eval benchmarks, especially some more complex tasks like QA, commonsense reasoning, etc.? To show this is a general methods that suits modern models and more challenging benchmarks
>
> Section 5.4 details the LM evaluation benchmarks we used which include QA and commonsense reasoning. Specifically, we have evaluated our English models on five QA benchmarks (PIQA, OpenBookQA, TriviaQA, BoolQ, and MathQA) and six commonsense reasoning tasks (HellaSwag, Winogrande, PIQA, OpenBookQA, ARC-Easy, and ARC-Challenge). Our German models were evaluated on two commonsense reasoning tasks (HellaSwag-DE and ARC-Challenge-DE) and a single questions answering task (TriviaQA). The German evaluation tasks are limited due to lack of availability. We are happy to include more evaluation results if there are benchmarks the reviewer believes are important. We would like to note, however, that “we use base models, that is our models have not been instruction-tuned, fine-tuned, or adapted to human preferences in any way”. The purpose of evaluating our models on LM evaluation benchmarks is exclusively to compare them with other models within our controlled study.

---

> ### Comment · Action_Editor_YCad · 2024-05-29
> **Please submit your official recommendation**
>
> Dear Reviewer:
>
> Thank you for reviewing the paper. Please submit your official recommendation for this paper as soon as possible, accounting for the authors responses (note that this is a separate step than submitting the review; see https://jmlr.org/tmlr/reviewer-guide.html). If there is a need for further discussion to arrive at an official recommendation, please raise it as soon as possible. If you have additional concerns, note that you can flag them as a comment in your official recommendation.
>
> Thank you!

---

### Review · Reviewer_Uarn · 2024-04-27

**Summary Of Contributions:**

This work challenges continual training of large language models by employing additional training data. The findings show that re-warming and re-decaying are necessary for better training with replay of prior data to avoid forgetting.

**Audience:**

No

**Broader Impact Concerns:**

None.

**Claims And Evidence:**

Yes

**Requested Changes:**

* I'd like to see the impact of using moment estimates in Adam optimizer to see if keep using the estimates in pre-training will be of benefit for the continual training.
* Also, better to analyze how the moment estimates will change during training.

**Strengths And Weaknesses:**

Strengths

* It is a simple yet effective approach in mitigating issues in continual training.

Weaknesses

* This work ignores the moment estimates in the Adam optimizer, and it is not clear how that will affect the effectiveness of training. Note that the instability issue might be coming from the inaccurate estimates in the early continual training, and simply preserving the estimates in the pre-training might solve all the issues.

---

> ### Author Response · Authors · 2024-05-03
> **General Comments and Reply to Requested Changes**
>
> Thank you for your careful reading of our paper and thoughtful feedback.  We are pleased that Uarm believes that our approach is “simple yet effective”.
>
> > This work ignores the moment estimates in the Adam optimizer, and it is not clear how that will affect the effectiveness of training. Note that the instability issue might be coming from the inaccurate estimates in the early continual training, and simply preserving the estimates in the pre-training might solve all the issues.
>
> We have provided a short experiment comparing the performance of keeping and dropping the optimizer states (see Section A.4 of the appendix). We would like to highlight that we explicitly made the decision to drop optimizer states in our experiments as they may not be available for many pre-trained models of interest (e.g. those on Huggingface hub). Moreover, state-of-the-art LLMs are trained using adamW with beta coefficients of 0.9 and 0.95 for the momentum and variance respectively. Therefore, any model using linear warmup will have essentially erased the contribution of momentum estimates from pre-training by the end of warmup. For instance, after 1000 steps of optimization, the moment estimates from pre-training will contribute to < 0.0044\% of the current moment estimates.
>
> **Reply to requested changes:**
>
> > I'd like to see the impact of using moment estimates in Adam optimizer to see if keep using the estimates in pre-training will be of benefit for the continual training.
>
> As mentioned above we have added an experiment to Section A.4 of the appendix. In our experiment, we observe no significant difference between preserving or dropping the optimizer states. Therefore, we conclude that, in this setting, keeping optimizer states does not affect continual pre-training so there is no benefit.
>
> > Also, better to analyze how the moment estimates will change during training.
>
> Thank you for the suggestion. We have included a short discussion of how the momentum estimates evolve during continual pre-training in Section A.4 of the appendix.

---

### Public Comment · ~Seng_Pei_Liew1 · 2024-06-19
**A question on Figure 10**

Dear Authors,

Congratulations on the paper acceptance!

I have a question on the infinite learning rate schedule studied in Section 7.
It is mentioned that the last checkpoint of the constant phase (or before cooldown) is used for continual pre-training under the infinite learning rate schedule:

```The checkpoint obtained at the end of this phase is the one one should resume from when continuing to pretrain on a new dataset.```

However, in Figure 10a, I do not see any line continued from the checkpoint where the constant phase ends when performing continual pre-training.
For example, the black and purple spikes appear to be continued from the end of the cooldown phase (at 100B tokens), but not from the end of the constant phase (at around 85B tokens).
This also does not seem to be an artifact of plotting mistakes; if we "shift" the spike to 85B tokens, the value of loss before and after the spike occurs does not match.

Could you clarify this?

---

> ### Author Response · Authors · 2024-06-21
> **Transitioning between datasets under infinite learning rate schedules**
>
> Thank you!
>
> Infinite learning rate schedules provide an alternative to cosine annealing for continual pre-training that enables a smooth transition between datasets. In Figure 10a, we train two checkpoints for each dataset: a checkpoint that continues the constant phase for all data in this dataset and a decayed checkpoint (e.g., phase 4). When transitioning to the new datasets, we select the former.

---

### Decision · Action_Editor_YCad · 2024-06-03

**Recommendation:** Accept as is

**Comment:**

The reviewers overall agree that the paper meets the criteria for acceptance. Thus, I have chosen to accept the paper as-is. However, the reviewers still have some lingering concerns about generality and novelty, so I have not recommended this paper for further certification.

**Audience:**

The issue of how to effectively adapt a large model to new data without retraining from scratch is increasingly salient, and the authors show some promising results in the models they considered. Thus, the majority of the reviewers agreed that the paper likely would be of interest to at least some individuals in the TMLR audience, though the limitations noted above may slightly reduce the scope of that interest. I concur with the assessment.

**Claims And Evidence:**

All reviewers stated that the claims were sufficiently accurate and supported by the evidence. The paper thoroughly supports its main claims (e.g., evaluating on zero/few-shot tasks as well as perplexity) and usefully expands the scope of analyses in follow-up experiments (e.g., considering infinite learning rate schedules). However, the reviewers did highlight several weaknesses with respect to the generality of the results (e.g. scope of models and datasets considered). The authors have addressed many of these concerns to some extent in their revisions in response to the reviewers. In preparing the final version, I would encourage them to keep the reviewers points in mind as they revise the limitations and related work. In particular, pointing to references that justify the choices the authors have made, and any more recent or concurrent works that successfully build upon these ideas, might help to increase the perceived generality of the work.